# Unsupervised Mismatch Localization in Cross-Modal Sequential Data with Application to Mispronunciations Localization

**Wei Wei** [†]
*National University of Singapore*

*wei.wei@u.nus.edu*

**Hengguan Huang** [†]
*National University of Singapore*

*huang.hengguan@u.nus.edu*

**Xiangming Gu**
*National University of Singapore*

*xiangming@u.nus.edu*

**Hao Wang**
*Rutgers University*

*hw488@cs.rutgers.edu*

**Ye Wang**
*National University of Singapore*

*wangye@comp.nus.edu.sg*

**Reviewed on OpenReview:** *https://openreview.net/forum?id=29V0xo7jKp*

## Abstract

Content mismatch usually occurs when data from one modality is translated to another, e.g. language learners producing mispronunciations (errors in speech) when reading a sentence (target text) aloud. However, most existing alignment algorithms assume that the content involved in the two modalities is perfectly matched, thus leading to difficulty in locating such mismatch between speech and text. In this work, we develop an unsupervised learning algorithm that can infer the relationship between content-mismatched cross-modal sequential data, especially for speech-text sequences. More specifically, we propose a hierarchical Bayesian deep learning model, dubbed mismatch localization variational autoencoder (ML-VAE), which decomposes the generative process of the speech into hierarchically structured latent variables, indicating the relationship between the two modalities. Training such a model is very challenging due to the discrete latent variables with complex dependencies involved. To address this challenge, we propose a novel and effective training procedure that alternates between estimating the hard assignments of the discrete latent variables over a specifically designed mismatch localization finite-state acceptor (ML-FSA) and updating the parameters of neural networks. In this work, we focus on the mismatch localization problem for speech and text, and our experimental results show that ML-VAE successfully locates the mismatch between text and speech, without the need for human annotations for model training [1].

## 1 Introduction

Sequential data is prevalent in daily life and usually comes in multiple modalities simultaneously, such as video with audio, speech with text, etc. Research problems about multi-modal sequential data processing have attracted great attention, especially on the alignment problem, such as aligning a video footage with

---

[†]These authors contributed equally to this work.
[1]Codes will be soon available at https://github.com/weiwei-ww/ML-VAE

actions (Song et al., 2016) or poses (Kundu et al., 2020), aligning speech with its text scripts (Chung et al., 2018), aligning audio with tags (Favory et al., 2021), etc.

The content mismatch could come from various aspects. For example, mispronounced words in speech or incorrect human annotation will cause mismatch between speech and text; actors not following scripts will cause mismatch between the video and the pre-scripted action list. Locating such content mismatch is an important task and has many potential applications. For example, locating the content mismatch between speech and text can help detect mispronunciations produced by the speaker, which is crucial for language learning.

In this work, we focus on the problem of mismatch localization between speech and text inputs. In other words, i.e., to locate the mispronunciations in the speech produced by a speaker. Such mispronunciation localization task plays an important role in real-world applications such as computer-aided language learning system, where the mispronunciation localization results can be used to provide multimodal feedback for the users. Most existing cross-modal alignment algorithms assume that data from a variety of modalities are matched to each other (Song et al., 2016; Kundu et al., 2020; Chung et al., 2018; Favory et al., 2021); this strong assumption leads to difficulty in locating the mismatch between the two modalities. In particular, recent speech-text alignment approaches (Kürzinger et al., 2020; McAuliffe et al., 2017) mostly work under the assumption that the speaker has correctly pronounced all the words and are therefore incapable of detecting mispronunciation. Although earlier studies Finke & Waibel (1997); Hazen (2006); Braunschweiler et al. (2010); Bell & Renals (2015) have considered the mismatch between speech and text, they require human-annotated speech to train an acoustic model. However, labeling such data is labor intensive and expensive. Similarly, traditional mispronunciation detection methods (Leung et al., 2019) also need to be trained on a large number of human-annotated speech samples from second language (L2) speakers, whose annotation process is even more time-consuming and requires professional linguists' support. Furthermore, these studies on mispronunciation detection can only detect which phonemes/words in the text input are mispronounced, without locating them in the speech.

To address these issues, we propose a hierarchical Bayesian deep learning model, dubbed mismatch localization variational autoencoder (ML-VAE), which aims at performing content mismatch localization between cross-modal sequential data without requiring any human annotation during the training stage. Our model is a hierarchically structured variational autoencoder (VAE) containing several hierarchically structured discrete latent variables. These latent variables describe the generative process of speech from L2 speakers and indicate the relationship between the two modalities.

One challenge for ML-VAE is that training such an architecture is very difficult due to the discrete latent variables with complex dependencies. To address this challenge, we propose a novel and effective training procedure that alternates between estimating the hard assignments of the discrete latent variables over a specifically designed finite-state automaton (FSA) and updating the parameters of neural networks.

The main contributions of our work include:

- We propose a hierarchical Bayesian deep learning model, ML-VAE, to address the problem of content mismatch localization from cross-modal sequential data.
- Our ML-VAE is the first method that bridges finite-state automata and variational autoencoders; this is achieved via our proposed mismatch localization finite-state acceptor (ML-FSA), which allows the ML-VAE to locate the mismatch by searching for the best path in ML-FSA.
- To address the challenge of inferring the latent discrete variables with complex dependencies involved in ML-VAE, we propose a novel and effective alternating inference and learning procedure.
- We apply ML-VAE to the mispronunciation localization task; experiments on a non-native English corpus to demonstrate ML-VAE's effectiveness in terms of unsupervisedly locating the mispronunciation segments in the speech.

## 2 Related Work

**Cross-Modal Sequential Data Alignment** There have been several studies on aligning sequential data from different modalities. For example, for video-action alignment, most existing work focuses on learning

spatio-temporal relations among the frames. Dwibedi et al. (2019) proposes a self-supervised learning approach named temporal cycle consistency learning to learn useful features for video action alignment. Liu et al. (2021) proposes to learn a normalized human pose feature to help perform the alignment task. Song et al. (2016) adopts an unsupervised way to align the actions in a video with its text description. In terms of speech-text alignment, a recent study by Kürzinger et al. (2020) proposes using the CTC output to perform speech-text alignment. However, these aforementioned studies assume a perfect match in the content of the cross-modal sequential data, making it impossible to locate the content mismatch from the data.

Speech-text alignment is usually referred to as the forced alignment (FA) task (McAuliffe et al., 2017) in the speech processing community. There is a fairly long history of research related to FA. Traditional method is to run the Viterbi decoding algorithm (Forney, 1973) on a Hidden-Markov-Model-based (HMM-based) acoustic model with a given text sequence. Even though several FA studies have considered the particular problem setting where the text is not perfectly matched to the speech, they require training an acoustic model using a large amount of human-annotated speech, which is laborious and costly. For instance, Finke & Waibel (1997); Moreno et al. (1998); Moreno & Alberti (2009); Bell & Renals (2015); Stan et al. (2012) require an acoustic model for text-to-speech alignment with a modified lattice, and Bordel et al. (2012) requires an acoustic model to obtain the recognized phoneme sequence. Therefore, these methods fail to handle our problem setting, where the speech data from L2 speakers is unlabelled.

**Variational Autoencoders and Bayesian Deep Learning** Variational autoencoder (VAE) (Kingma & Welling, 2013) is proposed to learn the latent representations of real-world data by introducing latent variables. It adopts an encoder to approximate the posterior distribution of the latent variable and and a decoder to model the data distribution. VAEs have a wide range of applications, such as data generation (Mescheder et al., 2017), representation learning (Oord et al., 2017), etc. As an extension to VAE to handle sequential data (e.g., speech), the variational recurrent neural network (VRNN) (Chung et al., 2015) introduces latent variables into the hidden states of a recurrent neural network (RNN), allowing the complex temporal dependency across time steps to be captured. Along a similar line of research, Johnson et al. (2016) proposes the structured VAE, which integrates the conditional random field-like structured probability graphical model with VAEs to capture the latent structures of video data. Factorized hierarchical variational autoencoder (FHVAE) (Hsu et al., 2017) improves upon VRNN and SVAE through introducing two dependent latent variables at different time scales, which enables the learning of disentangled and interpretable representations from sequential data. However, the hierarchical models mentioned above are trained by directly optimizing the evidence lower bound (ELBO), which may fail when discrete latent variables with complex dependencies are involved. To address this issue, we propose a novel learning procedure to optimize our ML-VAE.

To deal with data from different modalities, Jo et al. (2019) proposes a cross-modal VAE to capture both intra-modal and cross-modal associations from input data. Theodoridis et al. (2020) proposes a VAE-based method to perform cross-modal alignment of latent spaces. However, existing work on processing cross-modal data focuses mainly on learning the relationship between modalities while ignoring the potential mismatch between them. Therefore, we propose the ML-VAE to address this issue. In general, our work belongs to the category of Bayesian deep learning (BDL) (Wang & Yeung, 2016; 2020; Wang et al., 2015; Huang et al., 2020; 2021; 2022), and is the first BDL method that performs mismatch localization.

## 3  Problem Formulation

Here, we propose to generalize the traditional speech-text alignment problem by considering the content mismatch between the input sequences (see Fig. 1). Specifically, given two cross-modal sequences: (1) a speech feature sequence $\mathcal{X} = (x_1, ..., x_T)$ ($x_t \in \mathbb{R}^{D_f}$, where $D_f$ is the feature dimension) as the source sequence, and (2) a phoneme sequence $\mathcal{C} = (c_1, ..., c_L)$ ($c_l \in \mathcal{P}$, where $\mathcal{P}$ is the phoneme set) as the target sequence, our goal is to identify the mismatched target elements of $\mathcal{C}$ (i.e., mispronounced phonemes) while locating the corresponding elements in the source sequence (i.e. incorrect speech segments). Concretely, let $\mathcal{C}' = (c'_1, ..., c'_L)$ be the mismatch-identified target sequence. We use the notation $c^*_l$ to present mismatched content; then $c'_l = c^*_l$ if $c'_l$ is identified as mismatched content (i.e., a mispronounced phoneme) and $c'_l = c_l$ if $c'_l$ is identified as matched content (i.e., a correctly pronounced phoneme). The final localization result $\hat{\mathcal{C}} = (\hat{c}_1, ..., \hat{c}_T)$ is therefore a repeated version of $\mathcal{C}'$ with each element $c'_l$ repeating for $d_l$ times, indicating that

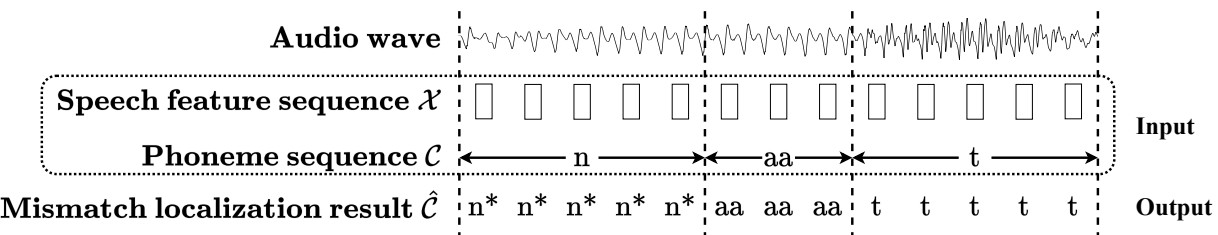

Figure 1: Problem formulation.

$c'_l$ lasts for $d_l$ frames. The mismatch localization performance is evaluated by averaging the intersection-over-union for each successfully detected mismatched phoneme, with more details to be introduced in Sec. 8.1.

## 4 Model

ML-VAE is a hierarchical Bayesian deep learning model (Wang & Yeung, 2020) based on VAE (Kingma & Welling, 2013). Our model aims at performing content mismatch localization when aligning cross-modal sequential data. In this work, we focus on the speech-text mismatch localization task.

This is made possible by decomposing the generative process of the speech into hierarchically structured latent variables, indicating the relationship between the two modalities.

ML-VAE is designed to use a hierarchical Gaussian mixture model (GMM) to model the match/mismatch between the sequences. Each Gaussian component corresponds to a type of mismatch, and the selection of the Gaussian component depends on the discrete latent variables of ML-VAE.

**Latent Variables and Generative Process** In the context of speech-text mismatch localization, since both the mismatch-identified phoneme sequence $\mathcal{C}'$ and the duration of each phoneme are unobservable, we introduce several latent variables to achieve the goal of mismatch localization for speech-text sequences. Instead of using a duration variable to describe the duration of each phoneme, we use a binary boundary variable sequence $\mathcal{B} = (b_1, ..., b_T)$, where $b_t = 1$ means the $t$-th frame marks the start of a phoneme segment, and thus $\sum_{t=1}^{T} b_t = L$. Besides, a binary correctness variable sequence $\Pi = (\pi_1, ..., \pi_T)$ is introduced to describe the matched/mismatched content in speech. Each entry $\pi_t \in \{0, 1\}$, with $\pi_t = 1$ if the $t$-th speech frame contains mismatched content (i.e. a mispronounced phoneme), and $\pi_t = 0$ otherwise.

To model the data generative process, at each time step $t$, we introduce three more latent variables: 1) $y_t$, which denotes the estimated phoneme, 2) $z_t$, which represents the Gaussian component indicator, and 3) $h_t$, which is the speech latent variable.

As shown in Fig. 2, we draw the estimated phoneme $y_t$ from a categorical distribution, and draw the correctness variable $\pi_t$ from a Bernoulli distribution. Meanwhile, the boundary variable $b_t$ is assumed to be drawn from a Bernoulli distribution $Bernoulli(\eta_t)$, which is further parameterized by a Beta distribution $Beta(\alpha, \beta)$.

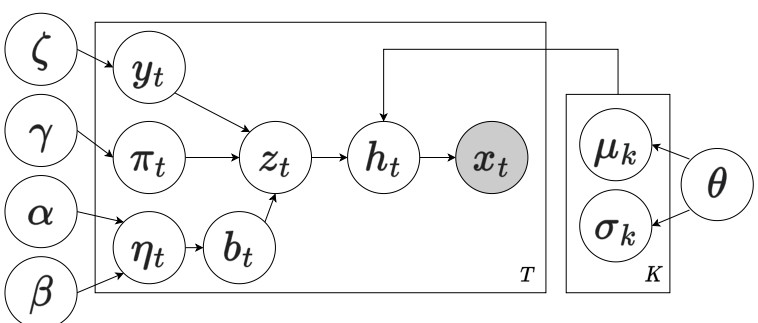

Figure 2: Graphical model for ML-VAE.

In the GMM part of ML-VAE, $z_t$, indicating the index of the selected Gaussian distribution, is an one-hot variable drawn from a categorical distribution, and $h_t$, the latent

variable of speech, is drawn from the selected Gaussian distribution. For each phoneme type, we use one component to model the matched content (i.e., correct pronunciation) and $N_m$ components to model different mismatch types (i.e., mispronunciation variants). Such a design is motivated by the following observations: all correct pronunciations of the same phoneme are usually similar to each other, while its mispronunciations often severely deviate from the correct one and have multiple variants (e.g., the consonant 'DH' may be mispronounced into 'T', 'D', or 'TH'). Therefore, for $N$ phoneme types, there are $N$ components modeling correct pronunciations and $N * N_m$ components modeling mispronunciations, making the GMM part of ML-VAE contain a total number of $K = N + N * N_m$ components.

Given the design introduced above, the generative process of ML-VAE is as follows:

- For the $t$-th frame ($t = 1, 2, \ldots, T$):
    - Draw the estimated phoneme $y_t \sim Categorical(\zeta)$.
    - Draw the boundary variable $b_t \sim Bernoulli(\gamma)$.
    - Draw the correctness variable $\pi_t \sim Bernoulli(\eta_t)$ , where $\eta_t \sim Beta(\alpha, \beta)$.
    - With $y_t$, $b_t$, and $\pi_t$, draw $z_t$ from $z_t|y_t, \pi_t, b_t \sim Categorical(f_z(y_t, b_t, \pi_t))$, where $f_z(\cdot)$ denotes a learnable neural network.
    - Given that $z_t[k] = 1$, select the $k$-th Gaussian distribution and draw $h_t|z_t \sim \mathcal{N}(\mu_k, \sigma_k^2)$.
    - Draw $x_t|h_t \sim \mathcal{N}(f_\mu(h_t), f_\sigma(h_t))$, where $f_\mu(\cdot)$ and $f_\sigma(\cdot)$ denote learnable neural networks.

The parameters of the prior distributions of the latent variables (e.g., $\alpha$, $\beta$) in ML-VAE are determined by the training data.

**An Example** We will take an audio recording of reading the word 'NOT' as an example to explain the generative process. More specifically, the phoneme sequence contains three phonemes: **n**, **aa**, and **t**. The estimated phoneme $y_t$ is sampled to estimate the phoneme pronounced by the speaker at each time step. The boundary variable $b_t$ is sampled to determine whether the speaker starts to pronounce the next phoneme at time step $t$ (e.g., from **n** to **aa**). The correctness variable $\pi_t$ is then sampled to determine whether this phoneme (e.g., **aa**) is correctly pronounced (e.g., whether **aa** is mispronounced as another phoneme). Based on $y_t$, $b_t$, and $\pi_t$, if the phoneme (e.g., **aa**) is correctly pronounced, the Gaussian component indicator $z_t$ will select the Gaussian distribution for the correct pronunciation to generate the speech latent variable $h_t$; otherwise, based on how the phoneme is mispronounced (e.g, **aa** mispronounced as **ae**), $z_t$ will select the corresponding Gaussian distribution to generate the speech latent variable $h_t$.

**Model Architecture** ML-VAE contains three components: boundary detector $\phi_b$, phoneme estimator $\phi_p$, and speech generator $\phi_h$, as shown in Fig. 3. The boundary detector outputs the approximated posterior $q_{\phi_b}(b_t|x_t)$, and the phoneme estimator outputs the approximated posterior $q_{\phi_p}(y_t|x_t)$. The speech generator aims to reconstruct the input speech feature sequence following the generation process discussed in Sec. 4. More details are introduced in Sec. 6.2.

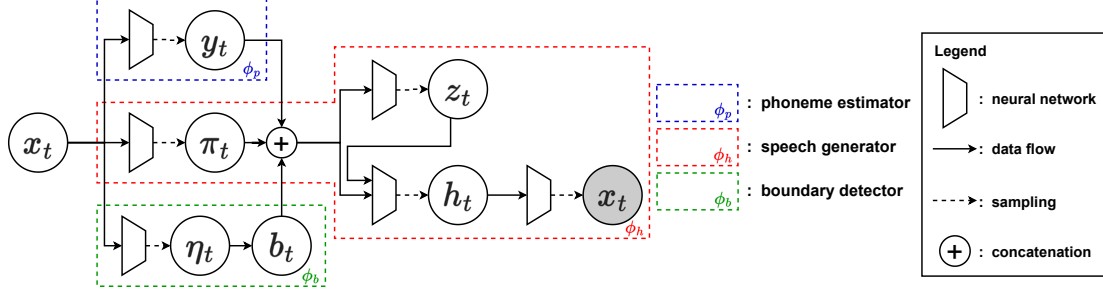

Figure 3: Model architecture for ML-VAE.

## 5 Mismatch Localization Finite-State Acceptor

In this section, we describe our proposed mismatch localization finite-state acceptor (ML-FSA) that bridges finite-state automata and variational autoencoders, allowing our ML-VAE to locate the mispronounced segment in the speech by efficiently searching the best path in ML-FSA. Our ML-FSA is a special type of finite-state acceptor (FSA) that describes possible hypothesis of mispronunciations and phoneme boundaries. We show ML-FSA for the $l$-th phoneme, $c_l$, in the given phoneme sequence in Fig. 4. Such a phoneme-level ML-FSA contains a set of states and a set of state-to-state transitions; the initial state 0 and the accepting state 5 are represented by a bold circle and concentric circles respectively; each transition includes a source state, a destination state, a label and a corresponding weight. Specifically, from the initial state (state 0), it can transit to state 1 or 3 based on pronunciation correctness ($R$ stands for correct pronunciation and $W$ stands for mispronunciation). This further leads to two different paths, one for correct pronunciation (denoted by $c_l$) leading to state 2, and the other one for mispronunciation (denoted by $c_l^*$) leading to state 4. In each path, at each time step, since each phoneme may last for several frames, it either still holds at the current state (denoted by $H$), or moves forward to the final state 5 (denoted by $S$).

With the help of the phoneme sequence $\mathcal{C}$, we can naturally build a sentence-level ML-FSA by combining the corresponding ML-FSA for each phoneme. With the sentence-level ML-FSA, a dynamic programming (DP) algorithm can be applied to search for the optimal path based on the weights on the transitions. More specifically, the weight of each transition in ML-FSA can be estimated by the modules in ML-VAE, with more details to be introduced in Sec. 6.1.

In contrast to weighted finite state transducers (WFSTs) Mohri et al. (2002), which only model the segmentation of audio into acoustic units (e.g., phonemes), our ML-FSA models the correctness of pronunciation ($R$ and $W$), the phoneme type ($c_l$, $c_l^*$), and the phoneme boundary ($H$ and $S$), which can jointly detect and segment the mispronunciation in the speech and thus perform *unsupervised content mismatch localization*. Furthermore, WFSTs require training a supervised acoustic model for weighting transitions between states, while our ML-

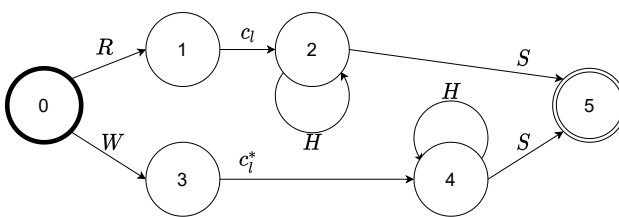

Figure 4: Proposed ML-FSA for the $l$-th phoneme, $c_l$.

FSA adopts the unsupervised ML-VAE to calculate all the weights of transitions. The details on locating the mispronunciation using ML-FSA are described in the next section. Note that ML-FSA takes all possible mispronunciation variants as a single symbol ($c_l^*$), which trades fidelity for efficiency. This is advantageous for addressing our problem settings: (1) our task focuses on distinguishing between the correct pronunciation and mispronunciation, rather than identifying the mispronunciation variants; (2) combining mispronunciation variants for each phoneme can dramatically reduce the computational cost of the dynamic programming (DP) algorithm searching for the optimal path.

## 6 Learning

The learning of ML-VAE consists of two main components: latent variables $\Psi = \{\mathcal{Y}, \mathcal{B}, \Pi\}$ and neural network parameters $\Phi = \{\phi_p, \phi_b, \phi_h\}$, where $\mathcal{Y} = (y_1, ..., y_t)$, and $\phi_p$, $\phi_b$, and $\phi_h$ denote parameters of the three modules of ML-VAE: phoneme estimator, boundary detector, and speech generator, respectively.

Following the traditional variational inference and the training objective for FHVAE (Hsu et al., 2017), the ELBO for the joint training objective of ML-VAE can be written as:

$$\text{ELBO} = \sum_{t=1}^{T} \Big( \mathbb{E}_{q(y_t, b_t, \pi_t | x_t)} \big[ \log p(x_t | y_t, b_t, \pi_t) \big] - D_{\text{KL}}(q(y_t, b_t, \pi_t | x_t) || p(y_t, b_t, \pi_t)) \Big), \tag{1}$$

where $D_{\text{KL}}$ is the function to calculate the Kullback–Leibler (KL) divergence, $q(y_t, b_t, \pi_t | x_t)$ is the joint approximate posterior distribution of the latent variables, $p(y_t, b_t, \pi_t)$ is the joint prior distribution.

However, unlike FHVAE, learning ML-VAE with such an objective function is very difficult due to ML-VAE's discrete latent variables with complex dependencies, as pointed out by Rolfe (2016). Furthermore, we empirically tested an intuitive approach – using a hard expectation-maximization (EM) algorithm (Moon, 1996) – and found that the system would not reliably converge. As pointed out by Locatello et al. (2019),

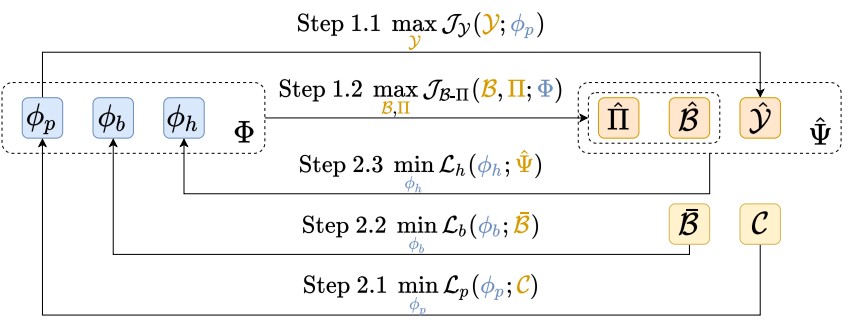

Figure 5: The training framework of ML-VAE.

lacking inductive bias for unsupervised training could render the model unidentifiable. Therefore, we improve this approach by providing pseudo-labels to some latent variables, as introduced by Khemakhem et al. (2020). Note that our learning algorithm remains unsupervised since it does not require any external data other than $\mathcal{X}$ and $\mathcal{C}$. Our new training procedure (see Fig. 5) can be described as:

- Step 1 (E step). Given the model parameters $\Phi$, estimate the hard assignments of the latent variables $\hat{\Psi} = \{\hat{\mathcal{Y}}, \hat{\mathcal{B}}, \hat{\Pi}\}$ by:

  - Step 1.1. Given $\phi_p$, estimate $\hat{\mathcal{Y}}$.
  - Step 1.2. Given $\Phi$, estimate $\hat{\mathcal{B}}$ and $\hat{\Pi}$.

- Step 2 (M step). Given the hard assignments of the latent variables $\hat{\Psi}$, optimize the model parameters by:

  - Step 2.1. Given the phoneme sequence $\mathcal{C}$ as the training target, optimize $\phi_p$ .
  - Step 2.2. Given a forced alignment result $\bar{\mathcal{B}}$, optimize $\phi_b$.
  - Step 2.3. Given the estimated hard assignments $\hat{\Psi}$ of latent variables, optimize $\phi_h$.

Our approach differs from the standard hard EM algorithm in several aspects. First, since the phoneme sequence $\mathcal{C}$ is given under our problem setting, the phoneme estimator $\phi_p$ can be trained by directly optimizing the cross-entropy loss towards $\mathcal{C}$ (Step 2.1). Second, we can directly adopt the predictions of $\phi_p$ to obtain $\hat{\mathcal{Y}}$ (Step 1.1). Third, we design an FSA to assist approximating a maximum a posteriori (MAP) estimate of $\hat{\mathcal{B}}$ and $\hat{\Pi}$ (Step 1.2). Fourth, we find that directly using $\hat{\mathcal{B}}$ from Step 1.2 deteriorates the model performance; therefore in Step 2.2, we adopt a forced alignment result $\bar{\mathcal{B}}$ (Tebelskis, 1995; McAuliffe et al., 2017) to train the boundary detector $\phi_b$. It is worth noting that obtaining such an alignment in Step 2.2 does not require any external data other than $\mathcal{X}$ and $\mathcal{C}$.

## 6.1 Step 1: Estimation of the Hard Assignments $\hat{\Psi}$

Next, we discuss the details of estimating the hard assignments.

**Estimation of $\hat{\mathcal{Y}}$** We obtain the estimated phoneme sequence $\hat{\mathcal{Y}}$ with:

$$\hat{\mathcal{Y}} = \underset{\mathcal{Y}}{\operatorname{argmax}} \, \mathcal{J}_{\mathcal{Y}}(\mathcal{Y}; \phi_p) = \underset{\mathcal{Y}}{\operatorname{argmax}} \, q_{\phi_p}(\mathcal{Y}|\mathcal{X}), \tag{2}$$

where $q_{\phi_p}(\mathcal{Y}|\mathcal{X})$ is the variational distribution calculated by the phoneme estimator $\phi_p$ to approximate the intractable true posterior $p(\mathcal{Y}|\mathcal{X})$.

**Estimation of $\hat{\mathcal{B}}$ and $\hat{\Pi}$ via ML-FSA** The estimation of $\hat{\mathcal{B}}$ and $\hat{\Pi}$ is done by finding the optimal path in the sentence-level ML-FSA. As introduced in Sec. 5, the weights of transitions of ML-FSA are all estimated by the model ML-VAE. More specifically, from state 0, the weights of the two transitions, labeled with $R$ and $W$, are estimated by the approximate posterior $q_{\phi_h}(\pi_t = 0|x_t, y_t, b_t)$ and $q_{\phi_h}(\pi_t = 1|x_t, y_t, b_t)$ respectively. Afterwards, the weights of the next two transitions, labeled with $c_l$ and $c_l^*$, are estimated by the posterior $q_{\phi_p}(y_t = c_l|x_t)$ and $q_{\phi_p}(y_t \neq c_l|x_t)$ respectively. Finally, the weights of the last two transitions, marked with $H$ and $S$, are estimated by the posterior $q_{\phi_b}(b_t = 0|x_t)$ and $q_{\phi_b}(b_t = 1|x_t)$ respectively. The details of how these posteriors are estimated will be introduced in Sec. 6.2.

Given all the weights in the sentence-level ML-FSA available, a typical dynamic programming (DP) algorithm can be used to find the optimal path which yields the maximum probability, whereby $\hat{\mathcal{B}}$ and $\hat{\Pi}$ are backtracked along the optimal DP path. Meanwhile, the optimal DP path also yields the mismatch localization result $\hat{\mathcal{C}}$. More details are given in Appendix A.

## 6.2 Step 2: Learning Model Parameters $\Phi$

In this section, we present how ML-VAE's parameters $\Phi = \{\phi_p, \phi_b, \phi_h\}$ are learned given the estimated $\hat{\Psi}$ obtained from the previous stage. Implementation and parameterization details can be found in Appendix E. Different objectives are proposed for $\phi_p$, $\phi_b$, and $\phi_h$.

### 6.2.1 Boundary Detector $\phi_b$

The boundary detector takes the speech feature sequence $\mathcal{X}$ as input and outputs the probability $q_{\phi_b}(b_t|x_t)$. The boundary variable $b_t$ is drawn from a Bernoulli distribution parameterized by an auxiliary latent variable $\eta_t$, that is, $b_t \sim \text{Bernoulli}(\eta_t)$, where we assume $\eta_t$ follows a Beta distribution: $\eta_t \sim Beta(\alpha, \beta)$. Therefore here we models an auxiliary distribution $q_{\phi_b}(\eta_t|x_t)$. As introduced, we use the forced alignment result sequence $\bar{\mathcal{B}} = (\bar{b}_1, ..., \bar{b}_T)$ as the pseudo-label to aid the training process. The training objective is to minimize the loss $\mathcal{L}_b$:

$$\mathcal{L}_b(\phi_b; \bar{\mathcal{B}}) = -\sum_{t=1}^{T} \Big( \mathbb{E}_{q_{\phi_b}(\eta_t|x_t)} \big[ \log p(b_t = \bar{b}_t|\eta_t) \big] - \lambda_b D_{\text{KL}}(q_{\phi_b}(\eta_t|x_t)||p(\eta_t)) \Big), \tag{3}$$

where $\lambda_b$ is a hyperparameter controlling the weight of the KL term, $q_{\phi_b}(\eta_t|x_t)$ is the approximate posterior distribution, and $p(\eta_t)$ is the prior distribution of $\eta_t$.

Note that no human annotation is required when obtaining the forced alignment result sequence $\bar{\mathcal{B}}$, since $\bar{\mathcal{B}}$ is obtained with $\mathcal{X}$ and $\mathcal{C}$ as inputs without using any human annotation of mispronounced phonemes in $\mathcal{C}$.

### 6.2.2 Phoneme Estimator $\phi_p$

Similar to the boundary detector, the phoneme estimator takes the speech feature sequence $\mathcal{X}$ as input and outputs the probability $q_{\phi_p}(y_t|x_t)$. We use the pseudo label $\tilde{\mathcal{C}} = (\tilde{c}_1, ..., \tilde{c}_T)$ as our training target, which can be obtained by extending the phoneme sequence $\mathcal{C}$ according to the estimated duration of each phoneme in $\bar{\mathcal{B}}$. Therefore, the model is optimized by minimizing the negative log-likelihood:

$$\mathcal{L}_p(\phi_p; \mathcal{C}) = -\sum_{t=1}^{T} \log(q_{\phi_p}(y_t = \tilde{c}_t|x_t)). \tag{4}$$

### 6.2.3 Speech Generator $\phi_h$

The speech generator $\phi_h$ aims to reconstruct the input speech feature sequence. It takes the speech feature sequence $\mathcal{X}$, along with the estimated values of the latent variables $\hat{\mathcal{Y}}$ and $\hat{\mathcal{B}}$ as input and reconstructs the speech feature sequence following the generation process discussed in Section 4.

We use the variational inference to learn the speech generator $\phi_h$ and thus the ELBO loss is calculated by:

$$\mathcal{L}_r(\phi_h) = -\sum_{t=1}^{T} \Big( \mathbb{E}_{q_{\phi_h}(h_t|x_t)} \big[ \log p_{\phi_h}(x_t|h_t) \big] - \lambda_r D_{\mathrm{KL}}(q_{\phi_h}(h_t|x_t) || p(h_t)) \Big) \tag{5}$$

where $\lambda_r$ controls the weight of the KL term, $q_{\phi_h}(h_t|x_t)$ is the approximate posterior distribution, and $p(h_t)$ is the prior distribution of $h_t$.

Besides, we also augment the ELBO using the estimated correctness variable sequence $\hat{\Pi}$ obtained from Step 1 (E Step) as a supervision signal and the new loss is written as:

$$\mathcal{L}_h(\phi_h; \hat{\Psi}) = \mathcal{L}_r(\phi_h) + \lambda_l \mathcal{L}_l(\phi_h; \hat{\Psi}), \tag{6}$$

where $\lambda_l$ is an importance weight; $\mathcal{L}_l(\phi_h; \hat{\Psi})$ is the negative log-likelihood loss, which is computed by:

$$\mathcal{L}_l(\phi_h; \hat{\Psi}) = -\sum_{t=1}^{T} \log q_{\phi_h}(\pi_t = \hat{\pi}_t | x_t, y_t, b_t), \tag{7}$$

where $q_{\phi_h}(\pi_t|x_t, y_t, b_t)$ is the approximate posterior distribution.

### 6.3 Overall Learning Algorithm

The overall algorithm to learn ML-VAE is shown in Algorithm 1.

---
**Algorithm 1** Learning ML-VAE
---
**Input:** Speech feature sequence $\mathcal{X}$, phoneme sequence $\mathcal{C}$
**Output:** Mismatch localization result $\hat{\mathcal{C}}$
 1: Initialize the model parameters $\phi_p$, $\phi_b$, and $\phi_h$.
 2: Obtain the forced alignment result $\bar{\mathcal{B}}$.
 3: **while** not converged **do**
 4:     Estimate $\hat{\mathcal{B}}$ and $\hat{\Pi}$ with ML-FSA.
 5:     Using Eq. 4, optimize $\phi_p$ with the phoneme sequence $\mathcal{C}$.
 6:     Using Eq. 3, with the help of $\bar{\mathcal{B}}$, optimize $\phi_b$.
 7:     Given $\hat{\Pi}$, optimize $\phi_h$ using Eq. 6.
 8: **end while**
 9: Obtain the mismatch localization result $\hat{\mathcal{C}}$ with ML-FSA.
10: **return** $\hat{\mathcal{C}}$
---

## 7 ML-VAE with REINFORCE Algorithm

In practice, we found that the training procedure is sometimes not stable due to large gradient variance. We suspect this is due to the sampling process of the discrete latent variables (such as $\pi_t$). To overcome such an issue and better reason about the distribution of the correctness variable during training, we further propose a variant of ML-VAE that uses the REINFORCE algorithm (Williams, 1992), which we call ML-VAE-RL.

Following the work by Xu et al. (2015), the reward term is defined as $\mathcal{R}(\Pi) = \mathcal{L}_h(\phi_h; \hat{\Psi})$, and to further reduce the gradient variance, a baseline term $b(\mathcal{X})$ is introduced to calculate the calibrated reward term $\hat{\mathcal{R}}(\Pi) = \mathcal{R}(\Pi) - b(\mathcal{X})$. $b(\mathcal{X})$ is estimated by a fully-connected neural network and trained by minimizing the mean squared error (MSE) between $b(\mathcal{X})$ and the uncalibrated reward $\mathcal{R}(\Pi)$: $\mathrm{MSE}(b(\mathcal{X}), \mathcal{R}(\Pi)) = \frac{1}{T} \sum_{t=1}^{T} [b(x_t) - \mathcal{R}(\pi_t)]^2$.

Then we define the new optimization objective $\mathcal{L}_{rl}(\phi_h; \hat{\Psi})$ whose gradient with respect to the model parameters $\phi_h$ can be calculated with the Monte Carlo method:

$$\nabla_{\phi_h} \mathcal{L}_{rl}(\phi_h; \hat{\Psi}) = \sum_{i=1}^{N_{mc}} \left( \nabla_{\phi_h} \mathcal{L}_h(\phi_h; \hat{\Psi}) + \hat{\mathcal{R}}(\Pi^{(i)}) \nabla_{\phi_h} \left[ -\log q_{\phi_h}(\Pi^{(i)} = \hat{\Pi} | \mathcal{X}, \mathcal{Y}, \mathcal{B}) \right] \right) - \nabla_{\phi_h} H[\Pi], \quad (8)$$

where $H[\pi]$ is the entropy of the distribution of the correctness variable whose gradient can be calculated explicitly. $N_{mc}$ is the number of Monte Carlo samples, and $\Pi^{(i)}$ is the $i$-th sample drawn from the posterior distribution $\Pi | \mathcal{X}, \mathcal{Y}, \mathcal{B}$.

With the help of the REINFORCE algorithm, Step 7 in Algorithm 1 is replaced by optimizing with Eq. 8 so that $\phi_h$ is reinforced to sample the correctness variable sequence $\Pi$ that produces lower $\mathcal{L}_h(\phi_h; \hat{\Psi})$.

## 8 Experiments

We evaluate our proposed ML-VAE and ML-VAE-RL on the mispronunciation localization task to test their mismatch localization ability. We first conduct experiments on a synthetic dataset (Mismatch-AudioMNIST) and then further apply our proposed models to a real-world speech-text dataset (L2-ARCTIC).

### 8.1 Evaluation Metrics

The F1 score is a commonly used metric to evaluate binary classification tasks, e.g., mispronunciation detection (Leung et al., 2019). However, we find that F1 score is not suitable to evaluate the mispronunciation localization task, as it is computed without evaluating if the model successfully locates the detected mispronunciations in speech.

As such, we improve upon the traditional F1 score by proposing a set of new evaluation metrics. For each true positive (TP) case, we calculate the intersection over union (IoU) metric of its corresponding phoneme segment, which demonstrates the performance of localization. IoU is computed by $IoU = \frac{I}{U}$, where $I$ and $U$ are the intersection length and the union length obtained by comparing the detected phoneme segment with the ground truth phoneme segment. Then such IoU metrics of all TP cases are summed and denoted as $\text{TP}_{\text{ML}}$. After calculating $\text{TP}_{\text{ML}}$, the TP in equations to calculate the F1 score is replaced with $\text{TP}_{\text{ML}}$ to calculate our proposed metrics. For example, if there are two TP cases detected by the model, then instead of calculating F1 score with TP $= 2$, we calculate the IoU of these two cases' corresponding phoneme segment, e.g. 0.3 and 0.6 respectively. Then we calculate our proposed metrics with $\text{TP}_{\text{ML}} = 0.3 + 0.6$. Our proposed metrics are denoted as mismatch localization precision ($\text{PR}_{\text{ML}}$), recall ($\text{RE}_{\text{ML}}$), and F1 score ($\text{F1}_{\text{ML}}$).

To be more specific, we first calculate the the intermediate metrics: true positive (TP), true negative (TN), false positive (FP), and false negative (FN). Then for all the TP cases detected by the model, we take the sum of their corresponding segments' intersection over union (IoU), which is denoted as $\text{TP}_{\text{ML}}$. Then the final metrics are computed as: $\text{PR}_{\text{ML}} = \frac{\text{TP}_{\text{ML}}}{\text{TP}+\text{FP}}$, $\text{RE}_{\text{ML}} = \frac{\text{TP}_{\text{ML}}}{\text{TP}+\text{FN}}$, $\text{F1}_{\text{ML}} = \frac{2 \times \text{PR}_{\text{ML}} \times \text{RE}_{\text{ML}}}{\text{PR}_{\text{ML}} + \text{RE}_{\text{ML}}}$

### 8.2 Baselines

To our knowledge, there is no existing work specifically designed for mispronunciation localization. Therefore, we adapt some existing methods on related tasks (e.g. ASR) to perform mispronunciation localization, and compare our proposed two ML-VAE models with them:

- **FA** (McAuliffe et al., 2017), which performs forced alignment using a deep-neural-network-HMM-based (DNN-HMM-based) acoustic model.

- **Two-Pass-FA** (Tebelskis, 1995), which first performs phoneme recognition based on a DNN-HMM-based acoustic model and then performs forced alignment on the recognized phoneme sequence and input speech.

- **w2v-CTC** (Baevski et al., 2020), which is built with wav2vec 2.0 and trained with CTC loss; CTC-segmentation (Kürzinger et al., 2020) is used to align the recognized phonemes with speech.

All models above are trained under the same problem setting where no human annotated data is used, i.e. only the speech feature sequence $\mathcal{X}$ and phoneme sequence $\mathcal{C}$ which contains mismatch are used during the training process.

### 8.3 Synthetic Dataset: Mismatch-AudioMNIST

We first test our proposed models on a synthetic dataset, named as **Mismatch-AudioMNIST**, which is built based on the AudioMNIST (Becker et al., 2018) dataset. Our synthetic dataset contains 3000 audio samples, each produced by concatenating three to seven spoken digits randomly selected from the original AudioMNIST. To simulate the content mismatch between audio and the corresponding text annotations, we randomly select 20.1% of the spoken digits as mismatched content, thereby labeling them as random digits. The total duration of Mismatch-AudioMNIST is 2.67 hours. The dataset is split into training, validation, and test sets by a 60:20:20 ratio, and the durations of the three sets are 96.3, 33.2, and 30.6 minutes respectively.

**Mispronunciation Localization Results** Table 1 shows the mispronunciation localization results of ML-VAE and ML-VAE-RL on the synthetic dataset Mismatch-AudioMNIST. We can see that all three baselines perform poorly for our mismatch localization task. More specifically, the vanilla forced alignment model (FA) fails to locate any mispronunciations due to its assumption that all the digits are correctly pronounced. Both of our proposed models (ML-VAE and ML-VAE-RL) are much superior to the baseline models, Two-Pass-FA and w2v-CTC, demonstrating the effectiveness of the proposed method in handling the unsupervised mismatch localization task. Note that in our problem settings, our training data contains mislabeled spoken digits and such mislabelling is unknown. This poses great challenges for acoustic model training. By further looking into the recognition results of Two-Pass-FA and w2v-CTC, we find that their recognition results from the acoustic model are very poor in terms of the mislabeled spoken digits, consequently limiting the baseline models' ability to detect such mislabels in the second stage processing.

Table 1: Mispronunciation localization results on Mismatch-AudioMNIST.

| Model | # Params | $PR_{ML}\%$ | $RE_{ML}\%$ | $F1_{ML}\%$ |
|---|---|---|---|---|
| FA | 3.3M | 0.00 | 0.00 | 0.00 |
| Two-Pass-FA | 3.3M | 6.22 | 2.43 | 2.28 |
| w2v-CTC | 352.5M | 1.72 | 3.84 | 2.30 |
| **ML-VAE (ours)** | 24M | 28.42 | 27.60 | 27.67 |
| **ML-VAE-RL (ours)** | 25M | **30.67** | **30.32** | **30.28** |

### 8.4 Real-World Dataset: L2-ARCTIC

We further apply ML-VAE and ML-VAE-RL to a real-world dataset: the **L2-ARCTIC** dataset (Zhao et al., 2018), which is a non-native English corpus containing 11026 utterances from 24 non-native speakers. Note that to evaluate mismatch localization, each annotated phoneme in the dataset needs to come with an onset and offset time label; to the best of our knowledge, the L2-ARCTIC dataset is currently *the only real-world dataset suitable for this problem setting.*

**Mispronunciation Localization Results** As shown in Table 2, similar to the results on Mismatch-AudioMNIST, Our ML-VAE models dramatically outperform Two-Pass-FA and w2v-CTC in all three metrics. Compared with original ML-VAE, the variant with REINFORCE algorithm yields better localization performance of mispronunciation.

Table 2: Mispronunciation localization results on L2-ARCTIC.

| Model | # Params | $PR_{ML}\%$ | $RE_{ML}\%$ | $F1_{ML}\%$ |
|---|---|---|---|---|
| FA | 3.3M | 0.00 | 0.00 | 0.00 |
| Two-Pass-FA | 3.3M | 0.64 | 0.96 | 0.77 |
| w2v-CTC | 352.5M | 1.29 | 2.26 | 1.64 |
| **ML-VAE (ours)** | 24M | 6.46 | 11.97 | 8.39 |
| **ML-VAE-RL (ours)** | 25M | **8.20** | **13.57** | **10.22** |

Interestingly, though w2v-CTC is based on a powerful wav2vec 2.0 acoustic model, which shows superior performance for ASR, it does not work well on the mispronunciation localization task; even with the largest number of parameters, it only slightly outperforms Two-Pass-FA, and underperforms our ML-VAE by a large margin. Our further analysis on w2v-CTC's output shows that such poor performance is mainly caused by the dataset which contains unknown content mismatches between the speech data and the corresponding text; w2v-CTC is fine-tuned on the speech data with mispronunciations, and therefore its recognition results tend to contain mispronunciations as well. For example, if a speaker always mispronounces the phoneme 'b' as 'd', a w2v-CTC model trained with such data tends to predict 'd', instead of 'b' during inference. Another reason for w2v-CTC's poor performance is that the CTC-segmentation algorithm assumes correct pronunciations and therefore does not work well on speech-text sequences containing mispronunciations.

Some more experiments are performed to demonstrate how traditional alignment methods fail on speech and text inputs with mispronunciations and such experimental results can be found in Appendix F.

**Ablation Study**   We present ablation study results to demonstrate the effectiveness of our proposed learning algorithm. Table 3 shows the results of ML-VAE-RL optimized with $\hat{\mathcal{B}}$ instead of $\bar{\mathcal{B}}$. The first row shows that directly using $\hat{\mathcal{B}}$ leads to very poor mispronunciation localization performance. The second row shows that our proposed optimization method effectively improves upon the traditional joint optimization method.

Table 3: Ablation study results.

| Model | $\text{PR}_{\textbf{ML}}\%$ | $\text{RE}_{\textbf{ML}}\%$ | $\text{F1}_{\textbf{ML}}\%$ |
|---|---|---|---|
| Ablation 1: ML-VAE-RL using $\hat{\mathcal{B}}$ instead of $\bar{\mathcal{B}}$ for optimization | 2.31 | 1.53 | 1.84 |
| Ablation 2: ML-VAE-RL w/ joint optimization | 4.90 | 2.89 | 3.64 |
| Ablation 3: ML-VAE-RL w/ $\hat{\mathcal{B}}$ & $\hat{\mathcal{Y}}$ estimated separately | 7.31 | 10.85 | 8.73 |
| **ML-VAE (ours)** | 6.46 | 11.97 | 8.39 |
| **ML-VAE-RL (ours)** | **8.20** | **13.57** | **10.22** |

The third ablation study compares our estimation method with a traditional one, which estimates $\hat{\mathcal{B}}$ and $\hat{\Pi}$ separately based on a single-path FSA. Details on this traditional algorithm can be found in Appendix B. Compared with this ablation model, our proposed ML-VAE-RL yields better results. This proves the effectiveness of our proposed estimation algorithm, which is specifically designed for the mismatch localization task.

# 9   Conclusions and Future Work

In this work, we present a hierarchical Bayesian deep learning model to address the mismatch localization problem in cross-modal sequential data. More specifically, two variants are proposed, namely ML-VAE and ML-VAE-RL. We also propose a learning algorithm to optimize our proposed ML-VAE. We focus on applying ML-VAE to the speech-text mismatch localization problem and propose a set of new metrics to evaluate the model's mispronunciation localization performance. Our experimental results show that it can achieve superior performance than existing methods, including the powerful wav2vec 2.0 acoustic model. We also perform ablation studies to verify the effectiveness of our training algorithm. One of the future research directions of this project is how to extend ML-FSA to address the case where multiple correct pronunciations exist.

# 10   Acknowledgement

We would like to thank our action editor Brian Kingsbury and the reviewers for their detailed and constructive comments and feedback. HW is partially supported by NSF Grant IIS-2127918 and an Amazon Faculty Research Award. This project is funded in part by a research grant A-0008153-00-00 from the Ministry of Education in Singapore.

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

# A    MAP estimation of $\hat{\mathcal{B}}$ and $\hat{\Pi}$

Given the sentence-level ML-FSA introduced in Sec. 5, the MAP estimate of $\hat{\mathcal{B}}$ and $\hat{\Pi}$ can be obtained by maximizing the following posterior distribution:

$$
\begin{aligned}
p(y|x) &\propto p(y)p(x|y) \\
&= \prod_{t=1}^{T} p(y_t|y_1,...,y_{t-1})p(x_t|y_t) \\
&= \prod_{t=1}^{T} p(y_t|y_1,...,y_{t-1})\frac{p(y_t|x_t)p(x_t)}{p(y_t)} \\
&\overset{a}{\approx} \prod_{t=1}^{T} p(y_t|y_1,...,y_{t-1})\frac{q_{\phi_p}(y_t|x_t)p(x_t)}{p(y_t)} \\
&\overset{b}{\propto} \prod_{t=1}^{T} p(y_t|y_1,...,y_{t-1})\frac{q_{\phi_p}(y_t|x_t)}{p(y_t)},
\end{aligned}
\tag{9}
$$

where in step $a$, $p(y_t|x_t)$ is approximated by $q_{\phi_p}(y_t|x_t)$. In step $b$, the constant term $p(x_t)$ is dropped. $p(y_t)$ is the prior of $y_t$, which is usually estimated with the frequencies of different phonemes in the training dataset. $q_{\phi_p}(y_t|x_t)$ is the approximate posterior, which can be computed using the phoneme estimator. $p(y_t|y_1,...,y_{t-1})$ is the transition probability which is determined by $p(b_t)$ and $p(\pi_t)$. Concretely, when calculating the transition probability $p(y_t|y_1,...,y_{t-1})$ given the sentence-level FSA, three different cases are considered: (1) the consecutive frames belong to the same segment; (2) the consecutive two frames belong to different segments, and the current segment *matches* the phoneme sequence; (3) the consecutive two frames belong to different segments, and the current segment contains content *mismatch* (i.e., a mispronounced phoneme). Corresponding to these three cases, we have:

$$
p(y_t|y_1,...,y_{t-1}) = \begin{cases} p(b_t=0), & \text{if } y_t = y_{t-1} \\ p(b_t=1)p(\pi_t=0), & \text{if } y_t \neq y_{t-1} \text{ and } y_t \in \mathcal{C} \\ p(b_t=1)p(\pi_t=1), & \text{if } y_t \neq y_{t-1} \text{ and } y_t \in \mathcal{C}^*, \end{cases}
\tag{10}
$$

where $\mathcal{C}^* = (c_1^*,...,c_L^*)$ denotes the mismatched elements in the phoneme sequence $\mathcal{C}$ (i.e. mispronounced phonemes)[2]. $p(\pi_t=0)$ and $p(\pi_t=1)$ are approximated by $q_{\phi_h}(\pi_t=0|x_t,y_t,b_t)$ and $q_{\phi_h}(\pi_t=1|x_t,y_t,b_t)$, which can be calculated using the speech generator. Similarly, $p(b_t=0)$ and $p(b_t=1)$ are approximated by $q_{\phi_b}(b_t=0|x_t)$ and $q_{\phi_b}(b_t=1|x_t)$, which can be obtained from the boundary detector.

Generally, the MAP estimation of $\hat{\mathcal{B}}$ and $\hat{\Pi}$ can be written as:

$$
\begin{aligned}
\hat{\mathcal{B}}, \hat{\Pi} &= \underset{\mathcal{B},\Pi}{\text{argmax}} \; \mathcal{J}_{\mathcal{B}\text{-}\Pi}(\mathcal{B},\Pi;\Phi) \\
&= \underset{\mathcal{B},\Pi}{\text{argmax}} \prod_{t=1}^{T} p(y_t|y_1,...,y_{t-1})\frac{q_{\phi_p}(y_t|x_t)}{p(y_t)}.
\end{aligned}
\tag{11}
$$

In practice, MAP estimation is achieved by following Eq. 11 and searching for the optimal path in ML-FSA, which can be solved by a DP algorithm.

---

[2]Note that strictly speaking, here we are approximating $p(y_t|y_1,...,y_{t-1})$ using $p(y_t|y_1,...,y_{t-1},c_1,...,c_t)$, where $c_1,...,c_t$ are the known canonical phonemes.

# B   Estimating $\hat{\mathcal{B}}$ and $\hat{\Pi}$ separately

In this section, we describe the algorithm which estimates the hard assignments for $\hat{\mathcal{B}}$ and $\hat{\Pi}$ separately. This algorithm is used to perform the ablation study.

## B.1   Estimation of $\hat{\mathcal{B}}$

Being different from our joint estimation algorithm discussed in the main paper, the boundary variable $\hat{\mathcal{B}}$ is first estimated without considering the correctness of the pronunciation.

When estimating $\hat{\mathcal{B}}$, we adopt the single-path FSA. Fig. 6 demonstrates the partial FSA for the $l$-th phoneme. From the initial state 0, there is only one path pointing to state 1, standing for the $l$-th phoneme, $c_l$. Then at each time step, since each phoneme may last for several frames, it either still holds at state 1 (denoted by $H$), or moves forward to the final state 2 (denoted by $S$). The sentence-level FSA can be constructed by connecting all phoneme-level FSA together, according to the input phoneme sequence $\mathcal{C}$.

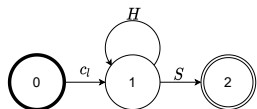

Figure 6: Single-path FSA.

Based on such an FSA, the weights on the transitions can be estimated by ML-VAE in a similar manner as described in Sec. 6.1. Then a DP algorithm can be used to find the optimal path in the sentence-level FSA, after which the estimated $\hat{\mathcal{B}}$ can be obtained by backtracking along the optimal path.

## B.2   Estimation of $\hat{\Pi}$

We then estimate $\hat{\Pi}$ given the $\hat{\mathcal{B}}$ obtained in the above procedure. If we are given the $l$-th phoneme segment which starts at $u$-th frame and ends at the $v$-th frame, we assume that all frames within this segment share the same pronunciation correctness label, that is, $\pi_u = \pi_{u+1} = \cdots = \pi_v = \pi'_l$, where $\pi'_l$ denotes the pronunciation correctness of the $l$-th phoneme.

Therefore, similar to Eq. 11, we can obtain the MAP estimation of $\pi'_l$ per segment:

$$\hat{\pi}'_l = \underset{\pi'_l}{\mathrm{argmax}} \prod_{t=u}^{v} p(y_t|y_1,...,y_{t-1}) \frac{q_{\phi_p}(y_t|x_t)}{p(y_t)} = \underset{\pi'_l}{\mathrm{argmax}} \begin{cases} p(\pi_s=0)\prod_{t=u}^{v} \frac{q_{\phi_p}(y_t=c_l|x_t)}{p(y_t=c_l)}, & \text{if } \pi'_l = 0 \\ p(\pi_s=1)\prod_{t=u}^{v} \frac{q_{\phi_p}(y_t\neq c_l|x_t)}{p(y_t\neq c_l)}, & \text{if } \pi'_l = 1, \end{cases} \quad (12)$$

where $q_{\phi_p}(y_t|x_t)$, $q_{\phi_p}(y_t=c_l|x_t)$, and $q_{\phi_p}(y_t\neq c_l|x_t)$ are the approximate posteriors. $y_t = c_l$ denotes that the $t$-th frame is correctly pronounced ($\pi'_l = 0$). $y_t \neq c_l$ denotes that the $t$-th frame is mispronounced ($\pi'_l = 1$), that is, $q_{\phi_p}(y_t \neq c_l|x_t) = 1 - q_{\phi_p}(y_t = c_l|x_t)$, and $p(y_t \neq c_l) = 1 - p(y_t = c_l)$.

# C   Case Study of ML-VAE

Besides quantitative analysis, we also provide a case study to showcase ML-VAE-RL in Fig. 7. Each rectangle in Fig. 7 represents one phoneme segment. The first and second rows show the input phoneme sequence and the actual phonemes pronounced by the speaker, respectively. The last row shows ML-VAE-RL's mispronunciation localization result. In the last row, a phoneme with a star sign (e.g. 't*') indicates that ML-VAE-RL detects a mispronounced phoneme (e.g. a mispronounced 't'). It is shown that ML-VAE successfully detects three out of four mispronunciations with a reasonable localization result.

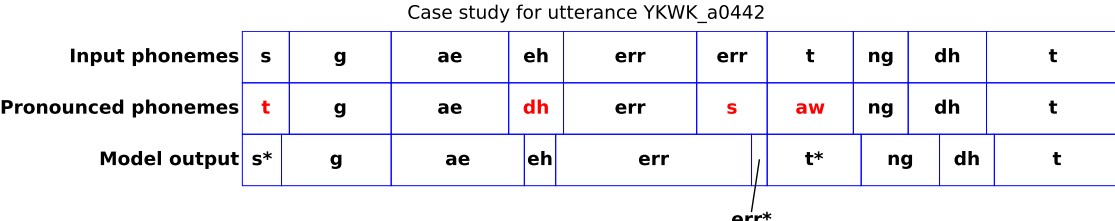

Figure 7: Case study for the last ten phonemes of the utterance YKWK_a0442. The first and second rows show the input phonemes and the actual phonemes pronounced by the speaker, respectively. The last row shows ML-VAE's predicted mispronunciation localization result.

## D  Case Study of Ablation Models

Fig. D shows the case study with the outputs of three ablation models presented in Table 3. It is shown that the joint optimization model yields the worst outputs because the poor alignment results. Compared with the other two ablation models, ML-VAE clearly outperforms them in terms of localizing the mispronounced phonemes.

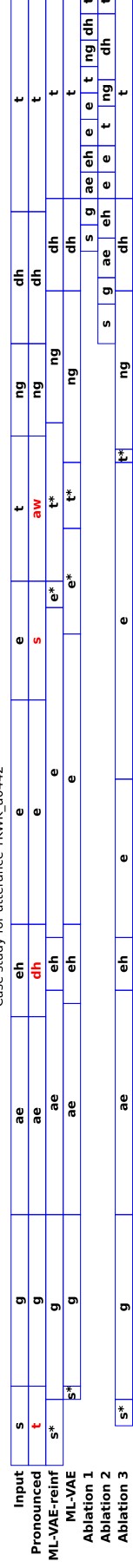

Figure 8: Case study for the last ten phonemes of the utterance YKWK_a0442. The first and second rows show the input phonemes and the actual phonemes pronounced by the speaker, respectively. The third row shows ML-VAE's predicted mispronunciation localization result. The last three rows show the outputs of the three ablation models as presented in Table 3. To save space, the phoneme **err** is denoted as **e** in this figure.

# E    Implementation Details

In this section, we introduce the implementation details of ML-VAE. The input feature sequence $\mathcal{X}$ is obtained by calculating the 40-dimension FBANK feature Young et al. (2002) from the speech signal. The phoneme sequence $\mathcal{C}$ is obtained from the text which is read by the speaker.

## E.1    Boundary Detector

The boundary detector includes two LSTM layers, each with 512 nodes, followed by two fully connected (FC) layers with 128 nodes and ReLU activations. They are followed by two separate output layers, each of which has only one node and a Softplus activation function. The output of these two layers is used to estimate the parameters $\alpha$ and $\beta$ to sample $v_t$, which is further used to sample $b_t$. The weight of the KL term $\lambda_b$ is set as 0.01.

## E.2    Phoneme Estimator

The phoneme estimator contains two LSTM layers; each LSTM layer has 512 nodes, followed by two FC layers, each with 128 nodes and ReLU. They are followed by a Softmax layer to give the estimation of the phoneme.

## E.3    Speech Generator

We adopt an encoder-decoder architecture to implement the speech generator. The encoder consists of three FC layers, each with 32 nodes, followed by four LSTM layers, each with 512 nodes. We use a FC layer with 512 nodes and ReLU to estimate the mean and variance of the Gaussian components. The decoder contains four bidirectional SRU layers (Lei et al., 2018), each with 512 nodes. They are followed by two FC layers, each with 120 nodes, to estimate the mean and variance of the data distribution. During training, $\lambda_r$ and $\lambda_l$ are set to 1 and 0.001, respectively.

**Gaussian Component Selection**    As described in the generative process, the Gaussian component indicator $z_t$ is sampled from $z_t|y_t, \pi_t, b_t \sim Categorical(f_z(y_t, b_t, \pi_t))$. In this section, we describe the implementation of $f_z(\cdot)$, which is defined as $f_z(y_t, b_t, \pi_t) = \text{softmax}(\rho_t * \epsilon + \delta_t)$, where $\rho_t$ is a scalar estimated by a two-layer multilayer perceptron (MLP) taking as inputs $y_t$, $b_t$, and $\pi_t$. Each layer of the MLP contains 128 nodes and a Sigmoid activation function; $\epsilon$ is a small constant, which is set as $1 \times 10^{-6}$ in our experiments; $\delta_t$ is an one-hot variable, i.e. $\delta_t[s] = 1$, and $s$ is computed by:

$$s = \begin{cases} (j-1) * (N_m + 1) + 1, & \text{if } \pi_t = 0 \\ (j-1) * (N_m + 1) + 1 + k, & \text{if } \pi_t = 1, \end{cases} \tag{13}$$

where $j$ denotes the phoneme label of $y_t$, i.e. $y_t[j] = 1$. In case that $y_t$ is mispronounced, $k \in [1, N_m]$ denotes the $k$-th mispronunciation variant of the phoneme $y_t$, which is implemented by a Gumbel Softmax function (Jang et al., 2016), i.e. $\tau_t = \text{Gumbel}(\text{NN}(x_t, y_t))$, where $\tau_t[k] = 1$ and NN is a simple neural network with three 128-node FC layers .

# F    Experimental Results of the Alignment Task

This section provides some experimental results of aligning speech and text inputs that contain mispronunciation, as shown in Table 4. Experiments are performed on the real-world dataset L2-ARCTIC. The intersection-over-union (IoU) is calculated for every phoneme in the input text and then averaged to evaluate the alignment performance. The experimental results show that the baselines, both the traditional forced alignment method and the CTC alignment algorithm, fail to yield reasonable alignment results, while our proposed ML-VAE and ML-VAE-RL can outperform both baselines in this task. Such results demonstrate the traditional alignment results would fail when there is mismatch in the inputs. For example, if there is

Table 4: Experimental results of aligning speech and text inputs with misproununciation on the L2-ARCTIC dataset.

| Model | Average IoU (%) |
|---|---|
| Two-Pass-FA | 4.68 |
| w2v-CTC | 5.73 |
| **ML-VAE (ours)** | 44.27 |
| **ML-VAE-RL (ours)** | **53.02** |

a mispronunciation in the middle of a sentence, it may cause errors in traditional alignment methods (e.g., forced alignment). Then such errors may accumulate and further degrade the alignment performance of the phonemes after the mispronounced one.

