# OpenReview forum: "Unsupervised Mismatch Localization in Cross-Modal Sequential Data with Application to Mispronunciations Localization"
_TMLR — Accepted by TMLR_

### Review · Reviewer_mR9q · 2022-10-25

**Summary Of Contributions:**

This work proposes a hierarchical Bayesian deep learning model, ML-VAE, to address the problem of content mismatch localization from cross-modal sequential data, which is the first method that bridges finite-state automata and variational autoencoders; this is achieved via our proposed mismatch localization finite-state acceptor (ML-FSA), which allows the ML-VAE to locate the mismatch by searching for the best path in ML-FSA. To address the challenge of inferring the latent discrete variables with complex dependencies involved in ML-VAE, this work proposes a novel and effective alternating inference and learning procedure. Furthermore, experiments on a non-native English corpus to demonstrate ML-VAE’s effectiveness in terms of unsupervisedly locating the mispronunciation segments in the speech.

**Audience:**

Yes

**Broader Impact Concerns:**

I have concerns about the following unlikely but seemingly possible ethical consequences:

(1) Could the audio datasets reveal the identities of the speakers? Do the speakers allow their speech to be studied and publicized?

(2) If employers use your techique to test interviewees' mispronunciation rate and accordingly decide whether to hire them (e.g. as a teacher/compere), could the interviewees whose accent lacks training samples be discriminated? For example, could your algorithm exaggerate their mispronunciation rate?

**Claims And Evidence:**

No

**Requested Changes:**

**Requested changes (critical for acceptance):** You may add clarifications for the following questions in the paper.

(1) About the list of ML-VAE generative process right before "**An Example**" in page 4:

(1.1) Should $h_k$ be $h_t$ in $f_{\mu}(h_k),f_{\sigma}(h_k)$?

(1.2) If $\pi_t=0$, it seems that $z_t$ can be uniquely determined by the correct phoneme $y_t$, yes? If yes, then it seems that $f_z(y_t,b_t,0)$ is not needed and we could simply keep $f_z(y_t,b_t)$ only for $\pi_t=1$. Clarification might be added to the paper. The rules in **Gaussian Component Selection** in Section 8.3 may also be accordingly adjusted.

(1.3) Is it more convenient to use $z_t\in\{1,2,\ldots,N\}$? If not, then you could clarify that $z_t\in\{0,1\}^{K}$ is an one-hot variable.

(1.4) What are the parameters of the 3 neural networks $f_z$, $f_{\mu}$ and $f_{\sigma}$? They seem to be missing in the learning process?

(2) Section 5 about ML-FSA:

(2.1) It seems that only one state transition occur from time $t$ to $t+1$, yes?

(2.2) What is the meaning of each state? In particular, what's the difference between states 1 VS 2, 3 VS 4? Once state 5 is reached, does it mean the end of $c_{\ell}$ and beginning (state 0) of $c_{\ell+1}$?

(3) In the training procedure in Section 6, How is the forced alignment result $\overline{\mathcal{B}}$ obtained? What's the difference between $\overline{\mathcal{B}}$ and $\hat{\mathcal{B}}$?

(4) Could you add derivation of the posterior of $\mathcal{B},\Pi$ in eq. (3)? Why are $b_t$ and $\pi_t$ missing in this posterior?

(5) For eq. (4), why not computing the probabilities of $b_t$ and $\pi_t$ exactly via $Categorical(\alpha)$ and $Bernoulli(\gamma)$? How can we ensure that these probabilities are close to posteriors $q_{\phi_b}$ and $q_{\phi_h}$ respectively? I suspect $p(y_t|y_1,\ldots,y_{t-1})$ may ought to be $p(y_t|y_1,\ldots,y_{t-1},x_1,\ldots,x_t)$ as (3) is to maximize posterior.

(6) I did not get the meanings of the parameters $\phi_p$, $\phi_b$ and $\phi_h$ until I saw the posterior notations $q_{\phi_p}(\mathcal{Y}|\mathcal{X})$, etc. from Section 6.1. Could you bring these notations to the beginning of Section 6 to more clearly introduce $\phi_p$, $\phi_b$ and $\phi_h$ as parameters of the neural networks describing posteriors $q_{\phi_p}(\mathcal{Y}|\mathcal{X})$, etc?

(7) How is the objective fucntion (5) obtained? Is it from ELBO in eq. (1)? If yes, should $b_t=\overline{b}_t$ be $x_t$ to match ELBO?

(8) In Section 7, in the objective function $\mathcal{L} _ {rl}$, should $\log(\Pi=\Pi^{(i)}|\mathcal{X},\mathcal{Y},\mathcal{B})$ be $\log q_{\phi_h}(\Pi=\Pi^{(i)}|\mathcal{X},\mathcal{Y},\mathcal{B})$? How is MSE defined (better to be written in formula if possible)?

(9) In Section 8.2, is the phoneme estimator network used to describe $q_{\phi_p}(y_t|x_t)$? What functions/posteriors are described by the encoder and decoder of the boundary detector and speech detector? You could write down notations like $q_{\phi_p}(y_t|x_t)$ to be clear.

(10) In Section 9.1, do you take each time frame $t$ as a sample? Do you take correct or incorrect pronunciation as positive? Could you define IoU metric, better to use equation if possible or cite a related work that contains the definition?

(11) In Section 9.2, I remember the FA method requires human anotation as said in the introduction, right? If yes, how to apply it with only $\mathcal{X}$ and $\mathcal{C}$? Also, do Two-Pass-FA and w2v-CTC require human anotation? You may clarify them in Section 9.2.

**Requested changes (not critical for acceptance):**

(1) In Section 3, how is the speech feature $x_t$ obtained, or what's the meaning of $x_t$? (e.g. volume, frequency, etc.) Also, I guess the phoneme sequence $\mathcal{C}$ is correct pronunication obtained from the text, yes? You might clarify these in the paper.

(2) Add "s" after "propose" and "adopt" in the first paragraph of Section 2. Do similar changes in the last two paragraphs of Section 2.

(3) In the third paragraph of Section 2, change "an decoder" to "a decoder".

(4) If possible, could you list some popular mismatch types in the second paragraph after Figure 2?

(5) In $\mathbb{E}$ of eq. (1), should $x$ be $x_t$?

**Strengths And Weaknesses:**

**Strengths:**
	This work solves an interesting and important application problem on locating the mismatch between text and speech. The proposed ML-VAE algorithm does not require perfect match assumption and human-annotation, but shows the highest accuracy in the experiments, which are solid advantages. The lit review and experiments look clear and comprehensive to me.

**Weaknesses:**
	There are many unclear and possibly incorrect points in the formulation of the proposed methods, as shown in the **requested changes (critical for acceptance)** below, so I answer "no" to "Claims And Evidence" until the authors address them.

---

> ### Author Response · Authors · 2022-11-10
> **Reply to Reviewer mR9q (1 of 2)**
>
> We thank Reviewer mR9q for the thorough feedback and suggestions to improve our paper.
>
> 1. > Should $h_k$ be $h_t$?
>
> 	Yes, it is a typo and has been corrected.
>
> 1. > If $\pi_t = 0$, it seems that $z_t$ can be uniquely determined by the correct phoneme $y_t$, yes? If yes, then it seems that $f_z(y_t, b_t, 0)$  is not needed and we could simply keep $f_z(y_t, b_t)$ only for $\pi_t = 1$. Clarification might be added to the paper. The rules in Gaussian Component Selection in Section 8.3 may also be accordingly adjusted.
>
> 	Yes, you are correct about the case when $\pi_t = 0$. However, we choose to integrate $\pi_t$ into $f_z$ so that we can have a unified Gaussian component selection process which makes the implementation step easier.
>
> 1. > Is it more convenient to use $z_t \in \{1, 2, \dots, N \}$?
>
> 	$z_t$ is indeed a one-hot vector and we have now put this into the revised paper. Actually, the dimension of $z_t$ depends on the value of both $N$ and $N_m$.
>
> 1. > What are the parameters of the 3 neural networks $f_z$, $f_\mu$ and $f_\sigma$? They seem to be missing in the learning process?
>
> 	Actually, the parameters of the three neural networks are included in the model parameters $\phi_b$, $\phi_p$, and $\phi_h$. We noticed that there is some confusion here. Therefore, we added a paragraph to the end of Section 4 "Model" along with a diagram explaining the model architecture and the meaning of the model parameters.
>
> 1. > I did not get the meanings of the parameters $\phi_p$, $\phi_b$ and $\phi_h$ until ...
>
> 	We apologize for the confusion caused. Hope the additional paragraph and the diagram at the end of Section 4 "Model" could clarify.
>
> 1. > It seems that only one state transition occur from time $t$ to $t + 1$, yes?
>
> 	For state 2 and 4, yes, at each time step $t$, there is only one state transition if it "holds" at this state. However, for other cases, there may be multiple transitions at one time step. Taking the first frame ($t = 0$) of a sentence as an example, if it is a correctly pronounced frame, then from time step $t = 0$ to $t = 1$, it transits from state 0 -> 1 -> 2, containing multiple transitions.
>
> 1. > What is the meaning of each state? In particular, what's the difference between states 1 VS 2, 3 VS 4?
>
> 	The states in the upper path present the correct pronunciation, while the states in the lower path present the mispronunciation. Then in each path, taking state 1 vs 2 as an example, at state 1, only the correctness is determined, and when transiting to state 2, the pronounced phoneme is also determined based on the input phoneme sequence $\mathcal C$.
>
> 1. > Once state 5 is reached, does it mean the end of $c_l$ and beginning (state 0) of $c_{l + 1}$?
>
> 	Yes correct. State 5 is the end of $c_l$ and also the initial state (state 0) of $c_{l + 1}$. This is also how the phoneme-level ML-FSA is used to build the sentence-level ML-FSA.
>
> 1. > In the training procedure in Section 6, How is the forced alignment result $\bar {\mathcal B}$ obtained?
>
> 	This question is about the forced alignment result $\bar{\mathcal B}$. As described in the problem formulation, there are two inputs of our model: $\mathcal X$ and $\mathcal C$. Therefore, the forced alignment result $\bar{\mathcal B}$ can be obtained directly with $\mathcal X$ and $\mathcal C$ without using any human annotation. Note that $\mathcal C$ can be obtained from the text that is read by the speaker; therefore, it does not need human annotation.
>
> 1. > What's the difference between $\bar {\mathcal B}$ and $\hat {\mathcal B}$?
>
> 	$\bar {\mathcal B}$ is the forced alignment result, which is used as a supervision signal for the boundary detector module. $\hat {\mathcal B}$ is the estimated hard assignment of the latent variable $\mathcal B$.
>
> 1. > Could you add derivation of the posterior of $\mathcal B, \Pi$ in eq. (3)? Why are $b_t$ and $\pi_t$ missing in this posterior? For eq. (4), why not computing the probabilities of $b_t$ and $\pi_t$ exactly via $Categorical(\alpha)$ and $Bernoulli(\gamma)$? How can we ensure that these probabilities are close to posteriors $q_{\phi_b}$ and $q_{\phi_h}$ respectively? I suspect ...
>
> 	We have revised the section describing how $\hat {\mathcal B}$ and $\hat\Pi$ are estimated by applying a dynamic programming algorithm on ML-FSA. Hope it clarifies.
>
> 1. >  How is the objective function (5) obtained? Is it from ELBO in eq. (1)? If yes, should $b_t = \bar b_t$ be $x_t$ to match ELBO?
>
> 	The objective function for loss ${\mathcal L}_b$ contains two terms. The first term is to utilize the forced alignment result as a supervision signal, while the second term is the KL term serving as a regularization term.

---

> ### Author Response · Authors · 2022-11-10
> **Reply to Reviewer mR9q (2 of 2)**
>
> 1. > In Section 7, in the objective function ${\mathcal L}_{rl}$, should $\log (\Pi = \Pi^{(i)}|\mathcal X, \mathcal Y, \mathcal B)$ be $\log q_\phi (\Pi = \Pi^{(i)}|\mathcal X, \mathcal Y, \mathcal B)$? How is MSE defined (better to be written in formula if possible)?
>
> 	Yes, there were some typos in the previous version of our paper. We have fully revised this section and added more details and discussions of the REINFORCE algorithm.
>
> 1. > In Section 8.2, is the phoneme estimator network used to describe $q_{\phi_p}(y_t | x_t)$? What functions/posteriors are described by the encoder and decoder of the boundary detector and speech detector? You could write down notations like $q_{\phi_p}(y_t | x_t)$ to be clear.
>
> 	Yes correct, the phoneme estimator network estimates the $q_{\phi_p}(y_t | x_t)$. We put the related content in Section 6.2 "Step 2: Learning Model Parameters $\Phi$", where the input and output of each module are described. To make things clearer, we added another paragraph to the end of Section 4 "Model" along with a diagram explaining the model architecture and parameters. Hope this could clarify.
>
> 1. > In Section 9.1, do you take each time frame $t$ as a sample? Do you take correct or incorrect pronunciation as positive? Could you define IoU metric, better to use equation if possible or cite a related work that contains the definition?
>
> 	Actually, the evaluation metrics are computed at phoneme level, i.e., each phoneme is considered as a sample. Under the problem setting of mispronunciation localization, the mispronunciation (incorrect pronunciation) is considered as positive.
>
> 	The IoU for each detected mispronunciation is computed by comparing the detected phoneme segment with the ground truth segment, then IoU is defined by $IoU = Intersection / Union$. For example, if the ground truth segment is from 0.1s to 0.2s, and the detected segment is from 0.05s to 0.15s, then we have $Intersection = 0.15 - 0.1 = 0.05s$ and $Union = 0.2 - 0.05 = 0.15s$, and $IoU = 0.05 / 0.15 = 1 / 3$. We have also revised this section trying to make things clearer.
>
> 1. > In Section 9.2, I remember the FA method requires human annotation as said in the introduction, right? If yes, how to apply it with only $\mathcal X$ and $\mathcal C$? Also, do Two-Pass-FA and w2v-CTC require human annotation?
>
> 	Yes, traditional FA works with speech and human-annotated text. However, in our work, we use only \mathcal X$ and $\mathcal C$ to obtain the forced alignment result to avoid using human annotation. Such a result is used as a supervision signal for the boundary detector.
>
> 	As for the two baselines, to make the experimental results comparable, they also work only with $\mathcal X$ and $\mathcal C$ as inputs (i.e., no human annotation for all baselines). We added such information to our revised paper.
>
> 1. > In Section 3, how is the speech feature $x_t$ obtained? I guess the phoneme sequence $\mathcal C$ is correct pronunciation obtained from the text, yes? You might clarify these in the paper.
>
> 	The feature is obtained by extracting the FBANK feature. Yes, $\mathcal C$ is the correct pronunciation obtained from the text. Thanks for your suggestions, and we have clarified them in the paper.
>
> 1. > Add "s" after "propose" and "adopt" in the first paragraph of Section 2. Do similar changes in the last two paragraphs of Section 2. In the third paragraph of Section 2, change "an decoder" to "a decoder".
>
> 	Thanks again for your detailed suggestions. We have revised the paper accordingly.
>
> 1. > If possible, could you list some popular mismatch types in the second paragraph after Figure 2?
>
> 	We added an example of a mismatch in that paragraph.
>
> 1. > In $\mathbb E$ of eq. (1), should $x$ be $x_t$?
>
> 	Yes, it is a typo and has now been corrected.
>
> 1. > Could the audio datasets reveal the identities of the speakers? Do the speakers allow their speech to be studied and publicized?
>
> 	Theoretically maybe. However, all the information that may reveal the speaker identity has all been removed (e.g., names), except for the audio signal. Before collecting such datasets, all the speakers will sign the agreement letter to allow their speech to be studied and published.
>
> 1. > If employers use your techique to test interviewees' mispronunciation rate and accordingly decide whether to hire them (e.g. as a teacher/compere), could the interviewees whose accent lacks training samples be discriminated? For example, could your algorithm exaggerate their mispronunciation rate?
>
> 	We believe that this question is about responsible, regulatable AI. Our paper is more about a new technological solution and it has many potential applications to benefit non-native speakers (e.g., computer-aided language learning systems). However, every coin has two sides, for example, nuclear fusion can be used to generate power to benefit people, but it can also be used to produce a nuclear bomb to destroy people. Therefore, rules and regulations are needed here.

---

> > ### Comment · Reviewer_mR9q · 2022-11-21
> > **Reviewer mR9q's 1st reply**
> >
> > Thank the authors for their elaborate answer and revision.
> >
> > Most of my questions are well clarified. Only my question (4) "Could you add derivation of the posterior of ..." remains unclear. Actually it asks about why the posterior is $\prod_{t=u}^v p(y_t|y_1,\ldots,y_{t-1})\frac{q_{\phi_p}(y_t|x_t)}{p(y_t)}$ in eq. (9) of the revised paper, while the authors' answer is about how to maximize this posterior.
> >
> > I found some other small non-critical suggestions when reading the revision, as listed below.
> >
> > Why do the 3 loss functions use different criteria in Section 6? For example, $\phi_h$ uses ELBO while $\phi_p$ uses negative log-likelihood. The reason could be explained in Section 6.
> >
> > The meanings of states 2 and 4 could be explicitly explained in Section 5.
> >
> > A citation may be added to FBANK feature, as that is unknown to readers outside this area like me.
> >
> > Typo: Near the end of page 4, change "anone-hotvariable" into "an one-hot variable".

---

> > > ### Author Response · Authors · 2022-11-25
> > > **Response to new comments**
> > >
> > > We are glad that you found our clarifications helpful, and thanks again for your further suggestions. Below is our response, which is also reflected in the revised paper.
> > >
> > > 1. Regarding your latest question on the posterior, we revised the paper by adding Appendix A to describe how it is obtained. Hope it may help.
> > >
> > > 1. > Why do the 3 loss functions use different criteria in Section 6?
> > >
> > > 	For the boundary detector $\phi_b$, we obtain the forced alignment result $\bar{\mathcal B}$ as a supervision signal. Therefore, the negative log-likelihood loss is used. For the speech generator $\phi_h$, the latent variable $h_t$ is not observable, and hence there is no annotation for training; in this case, we use the variational inference method for optimization. Meanwhile, we have the estimated value $\hat \pi_t$ for supervision; therefore the negative log-likelihood loss is used for ${\mathcal L}_l$.
> > >
> > > 1. > The meanings of states 2 and 4 could be explicitly explained in Section 5.
> > >
> > > 	This is a good suggestion. We have revised Section 5 to explain states 2 and 4 accordingly.
> > >
> > > 1. > A citation may be added to FBANK feature, as that is unknown to readers outside this area like me. Typo: Near the end of page 4, change "anone-hotvariable" into "an one-hot variable".
> > >
> > >  	Thanks for pointing them out. We have fixed them in the revised paper.

---

### Review · Reviewer_NADY · 2022-10-25

**Summary Of Contributions:**

This paper focuses on the problem of jointly aligning speech and text while appropriately locating mispronunciations in an unsupervised manner with only access to the speech sequence and the text phoneme sequence. The survey of related work presented in the paper states that there are existing techniques to align speech and text or to identify mispronunciations but most require human-annotated speech. There are no existing schemes for labeling the mispronunciations in the alignment without human-annotated examples.

This paper briefly describes the problem formulation and then proposes a hierarchical model based on the Variational Auto-Encoders (VAE) which maintains discrete latent variables corresponding to (i) the alignment of the speech and phoneme sequences, (ii) the correctness of the pronunciations, and (iii) the form of the mispronunciations, while continuous latent variables model the speech encoding-decoding. Given the discrete latent variables, the paper proposes the use of a finite-state acceptor and a decomposed learning/optimization scheme where the different components are optimized alternately while keeping the others fixed, and the finite-state acceptor is used to optimize for the alignment and correctness variables. The paper presents the specific decomposed optimization problems that need to be solved iteratively. In addition to the above scheme, a reinforcement learning based loss function is also presented and used.


After presenting the architecture for the different components in the hierarchical model, the proposed scheme is evaluated against some baselines adapted to this problem of jointly aligning and locating mispronunciations and is shown to have significantly improved performance over the baselines for a problem-specific modified evaluation metric. Beyond quantitative comparisons and additional ablation studies, the paper also presents a qualitative example demonstrating how the proposed scheme is able to align and locate mispronunciations simultaneously.


**Audience:**

Yes

**Broader Impact Concerns:**

There is a potential broader impact for the considered problem where "mispronunciation identification" for non-native speakers might be used in some way to disadvantage non-native speakers. Again, this concern stems from my lack of motivation for the problem and can be clarified by the authors. Another related broader impact pertains to how the canonical pronunciation is chosen since they can be different in different regions and still be "correct" and sufficient for communication. A model trained with data from one region might incorrectly identify and localize mismatches for speech from a different region.


**Claims And Evidence:**

Yes

**Requested Changes:**

There are three main adjustments to the submission which would secure my recommendation:

- Motivation for the need for joint alignment and  mispronunciation localization beyond just being able to align in a pronunciation-robust manner, and highlighting how existing alignment schemes (without mispronunciation localization) struggle with pronunciations while aligning.
- Addressing the presentation and technical clarity issues mentioned above (definitely including how the forced alignment in the proposed scheme is obtained without the requirement of human-annotated speech mentioned in Sec 2).
- Better explaining and motivating the baselines considered (such as how they were adapted) and highlight why they fail (this is partially done but the explanation is not clear; see above clarity comments/questions).


**Strengths And Weaknesses:**

**Strengths**


- This paper is attempting to solve a problem not previously solved in literature without any human-annotated examples in a unsupervised manner. Being able to solve such a hard problem of alignment and mistake localization in a unsupervised manner can be very useful.
- The proposed scheme of learning a hierarchical VAE with discrete latent variables using a finite-state acceptor can have applications even beyond this specific problem.
- The empirical performance of the proposed scheme relative to the baselines demonstrates very significant empirical improvements.



**Weaknesses**

- In my opinion, various technical details need better presentation for the reader to follow the proposed scheme. For example,
  - The Problem formulation (sec 3) can use (i) a conceptual/mathematical description of the data (and all relevant information like size of phoneme set, etc) available for learning, (ii) evaluation of the quality of a particular alignment to the ground-truth alignment (discussing the novel $\langle x \rangle_{ML}$ metrics more elaborately here). Also, given that the same term can be pronounced (correctly) with speech sequences of different length, how is that handled in the problem formulation (or why it is a non-issue)? What are the domains of the $x_i, i = 1, \ldots, T$ and $c_j, j = 1, \ldots, L$?
  - In Model (sec 4), it would be useful to discuss domain of $y_t, z_t, h_t$ and how they are mapped to/from the domains of $x_i, c_j$. How does the ML-VAE generative process to create $x_t$ depend on the input $c_j$? Is $y_t$ sampled from the input $\mathcal{C}$?
  - In ML-VAE with Reinforcement Learning (sec 7), it would be better to first highlight the challenges in Algorithm 1 and how a RL based training would allow us to circumvent those challenges.
  - Further examples discussed below.
- The discussion of related work states that forced alignment (FA) generally requires an acoustic model trained with human-annotated speech. In that case, it is not clear how the proposed scheme is able to obtain a forced alignment $\bar{\mathcal{B}}$ in Algorithm 1? The ablation study highlights the need for this FA so this part needs more explicit discussion for the reader to truly understand how the proposed scheme is able to get around this challenge.
- It is not clear why mismatch localization is critical in the alignment problem -- if the input speech and phenome sequences are appropriately aligned (even though some phenomes are mispronounced in the speech sequence), it is not clear why this is not sufficient and what might be some applications where we need to identify the mispronunciations in the speech. With mispronunciation, there might still exist a perfect match between speech and phoneme sequences but the speech part is non-canonical; it is not clear why matching non-canonical speech perfectly to phoneme is insufficient. This needs significant motivation in my opinion. Mispronunciation-robust alignment is definitely very important but requiring the localization of the mispronunciation needs motivation. Moreover, focusing on mispronunciations might be tricky since the "canonical" pronunciation is often regional and many terms have multiple canonical pronunciations even with native speakers.
- On a related note, it is not clear if existing alignment schemes are unsuccessful (in alignment only) if the problem contains mispronunciations or if they are able to perform the alignment without localizing mispronunciations. The presented metrics $PR_{ML}$ etc do not show that the baselines struggle with the alignment problem, and it might have been better to also report scores that evaluate the alignment. It is not clear that the mispronunciations are the issue and that the baselines are struggling with the alignment because of the mispronunciations.
- The Mismatch-AudioMNIST problem appears to be somewhat of a different task than the mispronunciation localization problem since, in this case, the mismatch pronunciation is **very very different** from the true pronunciation -- they are different numbers -- while in general speech-phoneme alignment problems the mispronunciations won't be very different from the true pronunciations. It would be great to get a better motivation as to why this is a useful problem to consider.



*Various specific clarity comments and questions*:

- Are their citations highlighting the failures of FHVAE, VRNN and SVAE with discrete latent variables? Are their empirical results in the paper that demonstrate this limitation?
- Can the Jo et al. (2019) and Theodoridis et al. (2020) perform alignment without mismatch localization (if there is a match)? Or are these schemes not meant for alignment at all?
- Is $N_m$ a hyper-parameter? How should it be set on a per-phoneme basis? How does this choice affect performance?
- On page 4, "Draw $x_t | h_t \sim \mathcal{N}(f_\mu (h_k), f_\sigma(h_k)$" should it be $h_t$ instead of $h_k$?
- Given the model description in Sec 4, the ML-FSA requires better introduction and contextualization. Also, what are the (transition) weights that need to be learned (so that we can apply DP to find the optimal path). Is this done on a phoneme level or sentence level (as mentioned that "With the help of [...] $\mathcal{C}$, we can [...] build a sentence-level ML-FSA ..."? Why process at phoneme level vs sentence level?
- In the learning (sec 6), where does the "prior distribution" $p(y_t, b_t, \pi_t)$ come from and how critical is its selection? How is prior $p(v_t)$ set? How are all the other priors specified?
- We need more clarity on the learnable parameters $\phi_p, \phi_b, \phi_h$ (in sec 6) and their connection to the model (described in sec 4) with $\alpha, \gamma, \mu, f_z, f_\mu, f_\sigma$ etc. There appears to be some mismatches in the notation.
- There are lots of unclear notation in sec 7 such as $\log(\Pi = \Pi^{(i)} | \mathcal{X}, \mathcal{Y}, \mathcal{B})$ and $MSE(\mathcal{R}(\Pi), b(\mathcal{X}))$.
- The RL loss seems very confusing where $\mathcal L_{h} (\phi_h, \hat{\Psi})$ is in the loss but is also in there as part of $R(\Pi)$, and $R(\Pi)$ is also in the $\mathcal{L}_{bl}$ term. Also, it is not clear what is meant by "baseline term estimated by a fully-connected neural network" for $b(\mathcal{X})$ since it is not clear what term is being estimated and how the estimation is being done.
- The RL variation seems to introduce unnecessary complexity. Also it is not clear if it is replacing the role of th ML-FSA and how learning in Sec 6 (figure 4) is modified when this RL loss is utilized. Is it just a change in Step 2.3 or does the whole process change?
- What is $\alpha$ and $\beta$ in sec 8.1? The boundary detector appears to be modeled with a $\gamma$ (sec 4) or a $v_t$ (sec 6.2.1) so it is not clear what is being done in 8.1.
- How is a Softplus (which is an elementwise activation function) converting the vector output of the decoder into a scalar in 8.1?
- It appears that very specific architectures are used for the different components of the model. It would be useful to motivate why such choices are made? Is it based on some existing architectures or is it a result of a architecture search? Also it would be good to understand how sensitive the performance is to the selected architectures (especially since we are counting number of parameters).
- The choice of the $f_z$ function and the related definition of $\delta_t$, $s$, $j, k$ appear to be quite involved and we do not see any intuition or justification behind these choices. Can this be better motivated or justified?
- Is $TP_{ML}$ same as $TP_{IoU}$? Are the TP computed based on the phoneme sequence or the final localization results?
- How are forced alignment baselines utilized ("adapted") for mismatch localization? How are the necessary acoustic models built without the human-annotated data (mentioned in sec 2)?
- In addition to ML-metrics, can we also look at the alignment scores? Like how well were the baselines and ML-VAE able to align $\mathcal{C}$ on top of $\mathcal{X}$?
- The example given for w2v-CTC would also be a challenging situation for the proposed ML-VAE, right. Since the learning process has no way to know that the pronounced 'd' should be a 'b'? How is the proposed scheme (or any scheme for that matter) able to connect a phoneme to the correct pronunciation without enough examples of the correct pronunciations.
- What is meant by "w2v-CTC is fine-tuned on speech data ..."? I thought all the learning was done with the training data provided? What is meant by fine-tuning here?

---

> ### Author Response · Authors · 2022-11-10
> **Reply to Reviewer NADY (1 of 3)**
>
> We thank Reviewer NADY for the detailed feedback and suggestions to improve our paper.
>
> 1. > The Problem formulation (sec 3) can use ...
>
> 	Thanks for your suggestions. We have revised the paper accordingly.
>
> 1. > Given that the same term can be pronounced (correctly) with speech sequences of different length, how is that handled in the problem formulation (or why it is a non-issue)?
>
> 	Yes indeed. A phoneme can be pronounced with different durations. Therefore, in the problem formulation, $\mathcal C '$ is different from $\hat{\mathcal C}$ that $\hat{\mathcal C}$ is a repeated version of $\mathcal C '$ with each phoneme lasting for a few frames. More specifically, in our model, the boundary variable $b_t$ is proposed to model the different durations of the phonemes.
>
> 1. > In Model (sec 4), it would be useful to discuss domain of $y_t, z_t, h_t$ and how they are mapped to/from the domains of $x_i, c_j$ . How does the ML-VAE generative process to create $x_t$ depend on the input $c_j$? Is $y_t$ sampled from the input $\mathcal C$?
>
> 	We have revised the paper accordingly to make it clearer. Besides, we also added another paragraph to the end of Section 4 "Model" along with a diagram explaining the model architecture. With the diagram, the data flow is presented more clearly. For example, $y_t$ is sampled from the posterior estimated by a neural network with $\mathcal X$ as input.
>
> 1. > We need more clarity on the learnable parameters $\phi_p, \phi_b, \phi_h$ (in sec 6) and their connection to the model.
>
> 	This is a question related to the previous one and it can be addressed by the new model architecture diagram.
>
> 1. > In ML-VAE with Reinforcement Learning (sec 7), it would be better to first highlight the challenges in Algorithm 1 and how a RL based training would allow us to circumvent those challenges.
>
> 	Thanks for the suggestions. We have fully revised this section and added more discussions of the REINFORCE algorithm.
>
> 1. > It is not clear how the proposed scheme is able to obtain a forced alignment $\bar{\mathcal B}$ in Algorithm 1? How are forced alignment baselines utilized ("adapted") for mismatch localization? How are the necessary acoustic models built without the human-annotated data (mentioned in sec 2)?
>
> 	This question is about the forced alignment result $\bar{\mathcal B}$. As described in the problem formulation, there are two inputs of our model: $\mathcal X$ and $\mathcal C$. Therefore, the forced alignment result $\bar{\mathcal B}$ can be obtained directly with $\mathcal X$ and $\mathcal C$ without using any human annotation. Note that $\mathcal C$  can be obtained from the text that is read by the speaker; therefore, it does not need human annotation.
>
> 1. > It is not clear why mismatch localization is critical in the alignment problem... It is not clear why this is not sufficient and what might be some applications where we need to identify the mispronunciations in the speech.
>
> 	Mispronunciation localization is a step in getting the computer to locate where the mispronunciation occurs in an utterance, and then in a word. Take for example the sentence “English is an interesting language”. In the real world, a listener would know exactly where a mispronunciation is, but this is not the same case with a computer, in which speech is represented as a continuous waveform. Computers don't have brains like humans to naturally tokenize recognized units in speech. Hence, we need to find engineering methods of modeling the mispronunciation detection process, as it is not obvious to a computer where a mispronunciation has occurred because computers don't have a knowledge representation of language the same way humans do. Furthermore, we are performing this task with unlabeled data, which means that we have no human-annotated indicators mapping the parts of the waveform that are mispronounced.
>
> 1. > With mispronunciation, there might still exist a perfect match between speech and phoneme sequences but the speech part is non-canonical; it is not clear why matching non-canonical speech perfectly to phoneme is insufficient...
>
> 	Since we are working with unlabeled data, it's difficult to locate mispronunciations just from the input speech and its force-aligned phoneme sequence without other information, like how the spectrogram signal of the "correct" phoneme looks compared to the wrong ones. Because we are doing unsupervised learning, we could literally be mapping the phonemes of "Mary had a little lamb" to the non-canonical speech of "Mary hella whistle brand" and have no way of representing why, how, or where this has gone wrong, at least in the case of an unsupervised computer model.

---

> > ### Comment · Reviewer_NADY · 2022-11-23
> > **Thank you for the response and revision (1/3)**
> >
> > Thank you for the responses to 1-6. They definitely address many of my clarity questions. The new **Model Architecture** discussion and the Figure 3 is very helpful to understand what is going on. However, it is unclear how the input phenome sequence $\mathcal{C} = \{ c_1, \ldots, c_L \}$ is utilized in this model architecture. There is definitely a dependence on $\mathcal{C}$ since it is an input and making its role explicit would be useful.
> >
> > Regarding 6, it might be worthwhile to present explicitly how we can get a FA $\bar{\mathcal{B}}$ given $\mathcal{X}$ and $\mathcal{C}$ (even in the appendix). Is this something standard? As a non-expert in speech-text alignment, this step is not obvious to me. It is also not clear how the FA $\bar{\mathcal{B}}$ is different from the DNN-HMM-based FA baseline by McAuliffe et al. (2017).
> >
> > Responses 7-8 definitely address my motivation questions and the explanations & examples are much appreciated.

---

> > > ### Author Response · Authors · 2022-11-25
> > > **Response to new comments (1/3)**
> > >
> > > We are glad that the new figure is helpful. Regarding your questions:
> > >
> > > 1. > it is unclear how the input phenome sequence $\mathcal C$ is utilized in this model architecture.
> > >
> > > 	The phoneme sequence $\mathcal C$ is not a direct input to the model, as shown in the Model Architecture figure. Instead, we use this phoneme sequence to produce the forced alignment result $\bar{\mathcal B}$ as a supervision signal to optimize the boundary detector.
> > >
> > > 1. > It might be worthwhile to present explicitly how we can get a FA $\bar{\mathcal B}$ given $\mathcal X$ and $\mathcal C$. Is this something standard?
> > >
> > > 	Yes, the forced alignment algorithm is a standard algorithm and is commonly used in hybrid DNN-HMM acoustic model training for automatic speech recognition (ASR). Therefore, we did not present the details in our paper. According to your suggestion, we added some related citations to the paper in case it is helpful for interested readers.
> > >
> > > 1. > It is also not clear how the FA $\bar{\mathcal B}$ is different from the DNN-HMM-based FA baseline by McAuliffe et al. (2017).
> > >
> > > 	We are sorry for the confusion. The FA result $\bar{\mathcal B}$ is obtained by the same algorithm as the one used in the FA baseline by McAuliffe et al. (2017). Note that although the forced alignment result contains errors and cannot reflect mispronounced phonemes, it serves as a supervision signal to optimize our ML-VAE’s boundary detector module.

---

> ### Author Response · Authors · 2022-11-10
> **Reply to Reviewer NADY (2 of 3)**
>
>
> 1. > It is not clear if existing alignment schemes are unsuccessful (in alignment only) if the problem contains mispronunciations or if they are able to perform the alignment without localizing mispronunciations. In addition to ML-metrics, can we also look at the alignment scores? Like how well were the baselines and ML-VAE able to align $\mathcal C$ on top of $\mathcal X$?
>
> 	These two are related questions. To answer these questions, some additional experimental results and discussions are provided in Appendix D "Experimental Results of the Alignment Task", demonstrating how traditional methods would fail on speech and text containing mispronunciations.
>
> 1. > Moreover, focusing on mispronunciations might be tricky since the "canonical" pronunciation is often regional and many terms have multiple canonical pronunciations even with native speakers.
>
> 	It is true that the standard accent varies. For example, General American is rhotic, Received Pronunciation is non-rhotic, Australian English is semi-rhotic, and Scottish Standard English has a notably different vowel system from the rest. However, these are, descriptively speaking, not mispronunciations. We assume that when creating a system, it has to be set to a target language and its standard accent. This is also why we only model some antiphones that are most definitely incorrect in the chosen standard, especially ones that form minimal pairs with each other in the standard. For example, in Received Pronunciation, the words "bed" and "bid" only differ by 1 vowel. This is a minimal pair, and swapping the vowels around would not be correct.
>
>
> 1. > The Mismatch-AudioMNIST problem appears to be somewhat of a different task than the mispronunciation localization problem... while in general speech-phoneme alignment problems the mispronunciations won't be very different from the true pronunciations.
>
> 	We agree with the fact that the audio mismatch task in Mismatch-AudioMNIST is very drastic, but we respectfully disagree that in general speech-alignment problems, the mispronunciations are not very different from true pronunciations, because they will at least differ in one of the three categories of place, manner, and articulation, in terms of consonants. Mispronunciation in vowels is indeed harder to justify since the vowel space is harder to quantify, but some are really very obvious in the case of language learners. Besides, in real-world cases, some language learners may make such "drastic" errors, such as mispronouncing "island" into "is-land". Furthermore, our experiments with the Mismatch-AudioMNIST dataset aim to demonstrate that our model can work not only at phoneme level, but also at word level. Therefore, we respectfully argue that Mismatch-AudioMNIST is useful for our work.
>
> 1. > Are their citations highlighting the failures of FHVAE, VRNN and SVAE with discrete latent variables? Are there empirical results in the paper that demonstrate this limitation?
>
> 	We added a citation of a related study to Section 6 of the revised paper.
>
>
> 1. > Can the Jo et al. (2019) and Theodoridis et al. (2020) perform alignment without mismatch localization (if there is a match)? Or are these schemes not meant for alignment at all?
>
> 	These two studies are proposed to capture the relationship among different modalities, not aiming at aligning two sequences. Furthermore, they are not proposed to detect and locate mismatches between two cross-modal sequences. Therefore, they are not applicable to our task.
>
> 1. > Is $N_m$ a hyper-parameter? How should it be set on a per-phoneme basis? How does this choice affect performance?
>
> 	Yes, $N_m$ is a hyper-parameter and it is set to be the same for all phonemes. In practice, we did a grid search of different values of $N_m$ and found out that $N_m = 3$ yields the best results.
>
> 1. > Should it be $h_t$ instead of $h_k$?
>
> 	Yes, it is a typo and has been corrected.
>
> 1. > Given the model description in Sec 4, the ML-FSA requires better introduction and contextualization.
>
> 	Thanks for your suggestion. We have revised the section and added some more details accordingly.
>
> 1. > In the learning (sec 6), where does the "prior distribution" $p(y_t, b_t, \pi_t)$ come from? How is the prior $p(v_t)$ set? How are all the other priors specified?
>
> 	$p(y_t, b_t, \pi_t)$ is the joint prior distribution of the latent variables and it is a part of the KL term in the ELBO equation. As for the prior $p(v_t)$, it is renamed to $p(\eta_t)$ in the revised paper, and it is set to be a Beta distribution $Beta(\alpha, \beta)$, where the values of $\alpha$ and $\beta$ are determined from the training data. Similarly, other priors are either determined from the training data (e.g., $y_t$), or a standard Gaussian distribution (e.g., $h_t$).

---

> > ### Comment · Reviewer_NADY · 2022-11-23
> > **Thank you for the response and revision (2/3)**
> >
> > Thank you for the new experiments in 1 regarding the challenges faced by the baselines with mispronunciations. The explanations in 2 and 3 are helpful in motivating the problem considered. Responses 4-9 address my questions and the corresponding updates in the revision are helpful.

---

> ### Author Response · Authors · 2022-11-10
> **Reply to Reviewer NADY (3 of 3)**
>
> 1. > There are lots of unclear notation in sec 7 such as ...
>
> 	We apologize for the unclear notations and we have revised them with correct equations and more detailed explanations.
>
> 1. > The RL loss seems very confusing... The RL variation seems to introduce unnecessary complexity.
>
> 	We apologize again for the confusion caused. As discussed above, we have fully revised this section and added more discussions of the REINFORCE algorithm. Hope this clarifies.
>
> 1. > What is $\alpha$ and $\beta$ in sec 8.1? The boundary detector appears to be modeled with a $\gamma$ (sec 4) or a $v_t$ (sec 6.2.1) so it is not clear what is being done in 8.1.
>
> 	There are some inconsistencies in our previous version of the paper. We have revised the related content introducing the boundary detector $\phi_b$ and the boundary variable $b_t$ to make things clearer.
>
> 	In the revised paper, $b_t \sim Bernoulli(\eta_t)$ and $\alpha$ and $\beta$ are the parameters of the prior of $\eta_t$.
>
> 1. > How is a Softplus converting the vector output of the decoder into a scalar in 8.1?
>
> 	We have revised the related content to make things correct. The final output layer has only one node so that it outputs a scalar, and then a Softplus activation function is used to map it to $(0, + \infty)$.
>
> 1. > Is $TP_{ML}$ same as $TP_{IoU}$? Are the TP computed based on the phoneme sequence or the final localization results?
>
> 	Yes, it is a typo and has now been corrected. TP is computed at phoneme level by comparing the localization results with the human-annotated ground truth.
>
> 1. > The example given for w2v-CTC would also be a challenging situation for the proposed ML-VAE, right? Since the learning process has no way to know that the pronounced 'd' should be a 'b'? How is the proposed scheme (or any scheme for that matter) able to connect a phoneme to the correct pronunciation without enough examples of the correct pronunciations.
>
> 	The hierarchical structure is proposed to address this issue. Suppose that we have some training samples of both correctly and incorrectly pronounced 'b'. Then correct pronunciations would be similar and form a single cluster, while mispronunciations will different and form different clusters (e.g., 'b' could be pronounced intl 'd', 't', etc). However, for traditional ASR models, they will be "confused" by such mispronounced samples.
>
>
> 1. > What is meant by "w2v-CTC is fine-tuned on speech data ..."? I thought all the learning was done with the training data provided? What is meant by fine-tuning here?
>
> 	Directly training w2v-CTC without pre-training on the L2-ARCTIC corpus gives quite poor performance. Therefore, we follow the conventional way of using the wav2vec model by fine-tuning a pre-trained wav2vec model on the training data which gives higher baseline results. Here, the "speech data" refers to the training L2 speech data in the dataset (e.g., the training set of L2-ARCTIC).
>
> 1. > There is a potential broader impact for the considered problem where "mispronunciation identification" for non-native speakers might be used in some way to disadvantage non-native speakers.
>
> 	We believe that this question is about responsible, regulatable AI. Our paper is more about a new technological solution and it has many potential applications to benefit non-native speakers (e.g., computer-aided language learning systems). However, every coin has two sides, for example, nuclear fusion can be used to generate power to benefit people, but it can also be used to produce a nuclear bomb to destroy people. Therefore, rules and regulations are needed here.

---

> > ### Comment · Reviewer_NADY · 2022-11-23
> > **Thank you for the response and revision (3/3)**
> >
> > Responses 1-2 and corresponding updates in the revision makes Section 7 clearer to me. Thank you for the clarifications in responses 3-4. Regarding 5, it seems that $TP_{ML}$ and $TP_{IoU}$ are separately defined and discussed in the latest revision without saying that they are the same, hence implying that they are different. The response 5 seems to imply $TP_{ML} = TP_{IoU}$  but in that case, might be good to just use one in the text.
> >
> > Thank you for the clarification on 6 and 7. Regarding 8, this makes sense. As reviewers, we are asked to identify potential broader impacts.

---

> > > ### Author Response · Authors · 2022-11-25
> > > **Response to new comments (2/3 and 3/3)**
> > >
> > > Sorry again for the confusion caused. Yes, you are correct that they are the same thing and we should use only one unified notation. We have revised the paper accordingly.

---

### Review · Reviewer_a3sc · 2022-10-26

**Summary Of Contributions:**

Most of the algorithms that align two modalities (e.g., text and speech) assume that the content of the two modalities are perfectly matched. This work investigates the case where there is a content-mismatch between cross-modal sequential data, by proposing an unsupervised learning algorithm — a hierarchical Bayesian deep learning model called ML-VAE (mismatch localization VAE).

Directly optimizing the ELBO is infeasible according to the authors, so they propose a new approach to optimize the ML-VAE. The training alternates between (1) estimating the "hard assignments of the discrete latent variables over a specifically designed finite-state automaton" (Section 6.1) and (2) updating the neural net parameters (Section 6.2).

The graphical model and generating process are demonstrated in Figure 2.

In addition, RL is also used, as a variant of the proposed algorithm.

The experiments are done on identifying mispronunciations in speech-text mismatch; the experiments are done on non-native English corpus. Qualitative analysis is done.

**Audience:**

Yes

**Broader Impact Concerns:**

I think it's worth discussing what specific real-world applications the authors' approach will benefit. Further, discuss the potential disadvantages to non-native speakers and how the authors can prevent misuse of the software.

**Claims And Evidence:**

Yes

**Requested Changes:**

- I think the authors should make it more clear in the title and the abstract that the proposed approach is only experimented on speech-text mismatch localization.
- Add motivation for only experimenting on mispronunciation alignment / why people would care about this task.
- A lot more discussion on RL (including motivation for integrating RL, how is baseline is trained, how samples are drawn, as well as potentially other RL algorithms).
- Carefully proofreading the equations (including page 4, Equation 3) and address my confusion above.

**Strengths And Weaknesses:**

The literature review in Section 1 and Section 2 is currently satisfactory to me.

Experimental results are encouraging.

Edit: The authors carefully integrated feedback into the revised draft.

--

It's not clear to me whether it's acceptable to only distinguish between correct pronunciation and mispronunciation, as discussed at the end of Section 5. For real-world application, we may need more complex variants, but it's unclear how a more complex version of modeling mispronunciation would work in practice, under the authors' framework.

The only task experimented in this paper is mispronunciation alignment, but it's not clear if it is clearly motivated. What would be the real use cases? Would there be other more impactful tasks for speech-text mismatch alignment that can be addressed by the authors' approach?

RL details are lacking. It would be more clear if the authors expand on the discussion on modeling baseline (the term that gets subtracted from reward) and how exactly it is learned. Small concern: Is there a reason the authors used naive REINFORCE instead of more recent approaches like PPO?
- Motivation of using RL is also lacking.
- The objective (ML-VAE + RL) has clear typos.

Some equations / notations are confusing...
- On page 4, what are the parameters for f's / how are they parametrized and learned?
- As discussed above, the ML-VAE + RL objective is confusing / has clear typos.
- Equation 3 is also confusing to me; it will be great if the authors expand the equation / make sure it is correctly written and clearly explained.

---

> ### Author Response · Authors · 2022-11-10
> **Reply to Reviewer a3sc (1 of 2)**
>
> We thank Reviewer a3sc for the detailed comments and suggestions to improve our paper.
>
> 1. > It's not clear to me whether it's acceptable to only distinguish between correct pronunciation and mispronunciation... For real-world application, we may need more complex variants, but it's unclear how a more complex version of modeling mispronunciation would work in practice...
>
> 	Actually, being able to make the binary decision if a phoneme is mispronounced and locate the mispronounced phoneme has many potential real-world applications, especially for computer-aided language learning systems. However, indeed, it is fair to say that in many cases, pronunciation evaluation is a spectrum rather than a binary. Our proposed framework is a starting point that tackles the mispronunciation localization problem in an unsupervised manner. In the future, it has the potential to be extended to model "more complex variants" by proposing a more complex model for modeling mispronunciations. For example, using anti-phones to model mispronunciation might give us some idea of place, manner, and articulation. A phone that fails to meet the ground truth requirements of all categories would be a mispronunciation. Failing in one category is not that bad, failing in two makes it a worse mispronunciation, and failing in all three is the worst result. Essentially, this information is helpful in deciding whether a pronunciation is good or not, and having a more complex version of modeling mispronunciation helps us extract this information.
>
>
> 1. > The only task experimented in this paper is mispronunciation alignment, but it's not clear if it is clearly motivated. What would be the real use cases? Would there be other more impactful tasks ...
>
> 	Mispronunciation localization is a step in getting the computer to locate where the mispronunciation occurs in an utterance, and then in a word. Take for example the sentence “English is an interesting language”. In the real world, a listener would know exactly where a mispronunciation is, but this is not the same case with a computer, in which speech is represented as a continuous waveform. Computers don't have brains like humans to naturally tokenize recognized units in speech. Hence, we need to find engineering methods of modeling the mispronunciation detection process, as it is not obvious to a computer where a mispronunciation has occurred because computers don't have a knowledge representation of language the same way humans do. Furthermore, we are performing this task with unlabeled data, which means that we have no human-annotated indicators mapping the parts of the waveform that are mispronounced.
>
> 	By addressing the mispronunciation localization task, one important application is the computer-aided language learning system, where the system can use the mispronunciation localization result to automatically provide the users with useful and also multimodal (e.g., text and audio playback) feedback. This is why localization is important because it provides a quantifiable knowledge representation to computers, which is valuable when we are trying to give informative feedback. For example, in a real-life situation, I might say that in the above utterance, the word "interesting" was mispronounced. I could even go on to describe whether the error was at the beginning, middle, or end of the word. Now, we are trying to automate this process in a computer system, hence the experiment of mispronunciation alignment.
>
> 1. > I think the authors should make it more clear in the title and the abstract that the proposed approach is only experimented on speech-text mismatch localization.
>
> 	Thanks for your suggestion. We agree that providing such information in the title and the abstract is important. We have revised the manuscript accordingly.
>
> 1. > On page 4, what are the parameters for f's / how are they parametrized and learned?
>
> 	The details of parameterization in the revised paper are presented in Appendix C "Implementation Details", where more details are introduced. For example, $f_\mu$ and $f_\sigma$ are both parameterized by four 512-node SRU layers and two FC layers with 120 nodes.
>
> 	Besides, we also added another paragraph to the end of Section 4 "Model" along with a diagram explaining the model architecture.

---

> > ### Comment · Reviewer_a3sc · 2022-11-22
> > **Feedback part 1**
> >
> > On #1: I see. It makes sense that the work is a starting point. The more complex model that is described makes a lot of sense.
> >
> > On #2: I see. Mismatch localization in an unsupervised manner is quite interesting. I wonder if the authors will do more experiments in the future (on different variants of English, or low-resource languages, for example).
> >
> > Regarding #4, the model architecture part at the end of Section 4 is very helpful. Thanks!
> >
> > I'm spending time going through other changes.

---

> > > ### Comment · Reviewer_a3sc · 2022-11-25
> > > **part 2**
> > >
> > > Thanks for the changes. The RL details were lacking but the newly added paragraphs on RL address my confusion.
> > >
> > > I also read through other reviewers' comments.
> > >
> > > A minor comment (no need to address): I understand that I've been asking quite a few questions on motivation. This may be out of scope for this paper, but I'm wondering whether people would use such an unsupervised model in the future. I think the unsupervised setting and getting it to work is really impressive. The setting that there's not enough human annotation is also very realistic. But would practitioners mostly rely on a few-shot setting in the future (i.e., learning based on a limited number of annotations)? Would a small number of annotations improve the performance, and by how much? I'm curious what authors think about this.

---

> > > > ### Author Response · Authors · 2022-11-25
> > > > **Reply to part 2**
> > > >
> > > > Thank you for your insightful comment on few-shot learning. Indeed, we expect that using few-shot learning on top of our unsupervised method would improve the performance of mismatch localization, compared to direct few-shot learning without our method. It is definitely interesting future work.

---

> > > ### Author Response · Authors · 2022-11-25
> > > **Reply to feedback part 1**
> > >
> > > Thanks for your feedback. We are glad that you found our response helpful. Regarding your question "```if the authors will do more experiments in the future```", the answer is affirmative; we are currently doing some more experiments to further apply our method to other datasets, such as other languages, by collecting data from language classes in our university. However, as mentioned in the paper, the annotation process (even for the test set) is very time-consuming and requires support from professional linguists. Therefore, it may take some time until we can have such a dataset ready for experiments.

---

> ### Author Response · Authors · 2022-11-11
> **Reply to Reviewer a3sc (2 of 2)**
>
> 1. > Equation 3 is also confusing to me...
>
> 	We apologize for the confusion caused. We have revised this section describing how $\hat {\mathcal B}$ and $\hat\Pi$ are estimated by applying a dynamic programming algorithm on ML-FSA. Hope it clarifies.
>
> 1. > RL details are lacking.
>
> 	We have fully revised the section introducing the ML-VAE-RL. More details on the REINFORCE algorithm have been added, including the motivation. The equations have also been revised.
>
> 1. > Further, discuss the potential disadvantages to non-native speakers and how the authors can prevent misuse of the software.
>
> 	We believe that this question is about responsible, regulatable AI. Our paper is more about a new technological solution and it has many potential applications to benefit non-native speakers (e.g., computer-aided language learning systems). However, every coin has two sides, for example, nuclear fusion can be used to generate power to benefit people, but it can also be used to produce a nuclear bomb to destroy people. Therefore, rules and regulations are needed here.

---

### Decision · Action_Editors · 2022-12-06

**Recommendation:** Accept as is

**Comment:**

The authors addressed most of the issues raised in the very detailed reviews provided by the three reviewers in their revision submitted on 10 Nov. 2022, and then on 25 Nov. 2022 they provided a second revision (after the reviewers had made their formal recommendations) that addresses remaining questions on derivation of the posterior and the use of the ground-truth phoneme sequences.

One question the action editor is left with is how this framework could be extended to handle the case where multiple pronunciations of a word are considered correct. I believe it would be possible to extend the ML-FSA, but in practice there might be challenges due to the presence of multiple correct paths through it.

**Audience:**

While all three reviewers selected "Yes" for the question of whether or not there is some portion of the TMLR audience that would be interested in the findings of the paper, reviewer a3sc expresses some concerns: "the specific application is quite niche in my opinion, and it's not clear whether the strictly unsupervised setting is necessary/realistic for future application."

The action editor is an expert in speech technologies, and I believe that this paper would be of great interest to members of the speech research community. There are practical cases in which the strictly unsupervised setting would be useful, especially for cases involving low-resource languages or other applications where there is little or no sufficiently representative annotated data available, and extension to the partially supervised case should be simple.

**Claims And Evidence:**

All three reviewers agree that the claims in the paper are supported by accurate, convincing and clear evidence, and after reading the latest revision of the paper, the action editor agrees.